# PLASTICITY-DRIVEN SPARSITY TRAINING FOR DEEP REINFORCEMENT LEARNING

## ABSTRACT

While the increasing complexity and model size of Deep Reinforcement Learning (DRL) networks promise potential for real-world applications, these same attributes can hinder deployment in scenarios that require efficient, low-latency models. The sparse-to-sparse training paradigm has gained traction in DRL for memory compression as it reduces peak memory usage and per-iteration computation. However, this approach may escalate the overall computational cost throughout the training process. Moreover, we establish a connection between sparsity and the loss of neural plasticity. Our findings indicate that the sparse-to-sparse training paradigm may compromise network plasticity early on due to an initially high degree of sparsity, potentially undermining policy performance. In this study, we present a novel sparse DRL training approach, building upon the naïve dense-to-sparse training method, i.e., iterative magnitude pruning, aimed to enhance network plasticity during sparse training. Our proposed approach, namely Plasticity-Driven Sparsity Training (PlaD), incorporates memory reset mechanisms to improve the consistency of the replay buffer, thereby enhancing network plasticity. Furthermore, it utilizes dynamic weight rescaling to mitigate the training instability that can arise from the interplay between sparse training and memory reset. We assess PlaD on various MuJoCo locomotion tasks. We assess PlaD on various MuJoCo locomotion tasks. Remarkably, it delivers performance on par with the dense model, even at sparsity levels exceeding 90%.

## 1 INTRODUCTION

Deep Reinforcement Learning (DRL) has witnessed substantial progress in recent years, with advancements spanning diverse domains such as protein structure prediction (Jumper et al., 2021), optimization of matrix multiplication algorithms (Fawzi et al., 2022), and the development of autonomous vehicles (Feng et al., 2023). While DRL harbors the potential to transform real-world applications via the utilization of increasingly complex and extensive networks, it concurrently poses substantial challenges. A key concern is the surge in model complexity, which is accompanied by significantly increasing computational load. This presents a notorious obstacle for the widespread deployment of DRL solutions, particularly for real-world applications that necessitate compact and efficient models, such as latency-constrained settings in controlling plasma (Degrave et al., 2022).

Sparse networks (or neural network pruning), since proposed by Mozer & Smolensky (1989); Janowsky (1989), have emerged as a prevalent technique for compressing model sizes, reducing memory demands, and shortening computational costs associated with modern neural network architectures. Numerous efforts have been made to incorporate sparse training in DRL. Specifically, Rusu et al. (2016); Schmitt et al. (2018); Zhang et al. (2019) utilize knowledge distillation to train a sparse student model. However, these approaches necessitate the pre-training or concurrent training of a dense model from which the final sparse DRL networks are distilled, adding to the complexity and computational burden. Sparse-to-sparse training techniques in supervised learning (Lee et al., 2019; Evci et al., 2020), which initialize with sparse networks, have garnered upsurging interest in the DRL field as the potential to restrict the peak memory cost and per-iteration computational FLOPs (in theory) (Arnob et al., 2021; Graesser et al., 2022; Tan et al., 2022; Grooten et al., 2023). For instance, Arnob et al. (2021) explore one-shot pruning before the start of training in offline RL domains, (Tan et al., 2022) propose a DST training method for online DRL with robust value learning

techniques, and (Graesser et al., 2022) perform systematic analysis on the effectiveness of different sparse learning algorithms in the online DRL setting.

However, sparse-to-sparse algorithms might take more iterations to coverage and achieve parity with the accuracy of dense training even under low pruning ratios (Liu & Wang, 2023), hence not always "cheaper" in terms of the total computation memory. For example, the training steps of RLx2 (Tan et al., 2022) ($3e6$) significantly exceed the required training steps in traditional dense training, i.e., $1e6$. Furthermore, we highlight an inherent increase in sparsity during dense training for DRL, a phenomenon that aligns with the loss of plasticity (Nikishin et al., 2022; Sokar et al., 2023), and subsequently potentially deteriorates policy performance. This observation calls into question the sparse-to-sparse training paradigm in DRL. Despite their dynamic nature, these methods enforce a high degree of fixed sparsity right from the start, which is associated with an immediate decrease in plasticity. Therefore, an interesting question remains open:

*Can we efficiently enhance plasticity in sparse DRL training to boost performance?*

In this paper, we present a novel dense-to-sparse training approach for DRL, named Plasticity-Driven Sparsity Training (PlaD). Specifically, PlaD initially aims to mitigate the loss of plasticity by periodically emptying the replay buffer, addressing the primary source of plasticity loss in DRL training, i.e., non-stationarity. Subsequently, PlaD introduces dynamic weight rescaling (DWR) to counteract the training instability induced by memory reset and sparse training process. Our approach is straightforward to implement and can readily be adapted to various pruning techniques. To illustrate the efficacy of enhancing plasticity in sparse DRL training, PlaD is built upon the simple yet effective iterative magnitude pruning (IMP) method (Han et al., 2015). The integration of these two novel components enhances network plasticity and training stability, enabling the policy performance on par with dense models under sparsity levels in excess of $90\%$. The primary contributions of this paper are as follows:

- We explore the inherent increase in sparsity during standard dense training in DRL and establish a link between sparsity training and the loss of plasticity within DRL.

- Inspired by these insights, we introduce PlaD, a plasticity-centric approach for sparse training within a dense-to-sparse training paradigm. The two innovative components of PlaD, namely memory reset and dynamic weight rescaling (DWR), necessarily enhance plasticity and stabilize the training.

- Through rigorous evaluation, we showcase the superior sparse training performance of PlaD when combined with a fundamental algorithm, i.e., SAC (Haarnoja et al., 2018), across several MuJoCo tasks (Todorov et al., 2012). Remarkably, even under one of the simplest pruning algorithms, i.e., IMP, PlaD achieves performance comparable to that of dense models, maintaining this standard even when the sparsity level surpasses $90\%$.

## 2 RELATED WORKS

### 2.1 SPARSE TRAINING

**Dense-to-Sparse Training.**   Dense-to-sparse training typically starts with a fully connected neural network (dense model/network), where weights are progressively or instantaneously reduced to zero, culminating in a sparse model (Zhu & Gupta, 2017; Gale et al., 2019; Louizos et al., 2018; You et al., 2019; Liu et al., 2020; Kusupati et al., 2020; Liu et al., 2021). Various techniques have been employed for the dense-to-sparse training paradigm, including random (Liu et al., 2019; 2022), magnitude (Han et al., 2015), $L_1$ or $L_2$ regularization (Wen et al., 2016; Louizos et al., 2018), dropout (Molchanov et al., 2017), and weight reparameterization (Schwarz et al., 2021). Standard post-training pruning can be considered a specific instance within this category, typically involving the complete pre-training of a dense network followed by multiple cycles of re-training, each incrementing the level of sparsity after pruning(Janowsky, 1989; Denton et al., 2014; Singh & Alistarh, 2020). Another stream of research is centered around the Lottery Ticket Hypothesis (LTH) (Frankle & Carbin, 2019; Chen et al., 2020). This hypothesis posits that a sparse "winning ticket" at initialization can be identified through an iterative process of training, pruning, and resetting. However, the peak per-iteration computational FLOPs in a dense-to-sparse training process can be as high as in full dense training.

Both post-pruning and LTH methods are known to be resource-intensive due to the necessity for multiple cycles of pruning and re-training.

**Sparse-to-Sparse Training.** Sparse-to-sparse training is designed to train an inherently sparse neural network from the outset and maintain the prescribed level of sparsity throughout the training process (Mocanu et al., 2016; Bellec et al., 2018; Liu et al., 2021). These approaches start with a sparse network prior to training. Some methodologies emphasize the dynamic change of topology evolution (Bellec et al., 2018; Mocanu et al., 2018; Mostafa & Wang, 2019; Evci et al., 2020), whereas others prioritize identifying a static sparse network before training (Lee et al., 2019; Wang et al., 2020; Tanaka et al., 2020). However, many of these algorithms, despite theoretically having lower peak per-iteration computational FLOPs, may require significantly more time to achieve performance comparable to that of dense-to-sparse training (Evci et al., 2020).

**Sparse Training in DRL.** Employing sparse training in DRL presents a greater challenge than in supervised learning due to inherent training instability and non-stationarity data streams (Evci et al., 2020; Sokar et al., 2021; Graesser et al., 2022). Drawing inspiration from knowledge distillation, Livne & Cohen (2020) train a sparse RL student network using iterative policy pruning based on a pre-trained dense teacher policy network. Similarly, Zhang et al. (2019) concurrently learn a smaller network for the behavior policy and a large dense target network. LTH has shown promise in DRL for identifying a sparse winning ticket via behavior cloning (Yu et al., 2020; Vischer et al., 2022). The sparse-to-sparse training paradigm has been adopted to mitigate the computational burden associated with policy distillation and dense-to-sparse training (Lee et al., 2021; Sokar et al., 2021; Arnob et al., 2021). To achieve sparse DRL agents, sparse-to-sparse training methods include block-circuit compression and pruning (Lee et al., 2021), sparse evolutionary training in topology evolution (Sokar et al., 2021), and one-shot pruning at initialization in offline RL (Arnob et al., 2021). A comprehensive investigation of various sparse-to-sparse training techniques applied to a variety of RL agents and environments is conducted by Graesser et al. (2022). Sparse DRL networks have also been found to enhance minimal task representation and filter noisy information (Vischer et al., 2022; Grooten et al., 2023). However, sparse-to-sparse training can potentially introduce high computational costs in terms of total training time to reach the optimal solution and may require complex strategies to stabilize training (Liu & Wang, 2023; Tan et al., 2022).

## 2.2 PLASTICITY OF NEURAL NETWORKS

The concept of neural network plasticity, which broadly refers to the capacity to adapt to new information, has recently garnered attention in the field of deep learning (Mozaffar et al., 2019; Berariu et al., 2021; Zilly, 2022). Emerging evidence indicates that managing the decline in neural network plasticity, particularly in the context of continuous learning with dynamic data streams, new tasks, and evolving environments, can lead to consistent performance enhancements throughout the training process (Achille et al., 2017; Ash & Adams, 2020; Igl et al., 2020; Dohare et al., 2021; Nikishin et al., 2022).

DRL is particularly susceptible to the effect of neural network plasticity due to the inherent non-stationarity in the targets and data flows (Nikishin et al., 2022; Igl et al., 2020; Sokar et al., 2023). Several techniques have been developed that focus on improving plasticity, and these have demonstrated remarkable performance. These techniques include controlling rank collapse (Kumar et al., 2021), periodically resetting the network (Nikishin et al., 2022; D'Oro et al., 2022; Schwarzer et al., 2023), reactivating dormant neurons (Sokar et al., 2023), imposing regularization on the initial network (Lyle et al., 2022), injecting randomly initialized layers (Nikishin et al., 2023), and layer normalization (Lyle et al., 2023).

## 3 PRELIMINARIES

We are interested in the standard RL formulation under the Markov Decision Process (MDP) formalism $\mathcal{M} = (\mathcal{S}, \mathcal{A}, \mathcal{R}, \mathcal{P}, \gamma)$. Usually, for one interaction process, the agent chooses an action $a \in \mathcal{A}$ based on the observed state $s \in \mathcal{S}$ from the environment, and then obtains a reward $r$ based on a reward function $r(s, a) : \mathcal{S} \times \mathcal{A} \rightarrow \mathbb{R}$. After getting the action $a$ from the agent, the environment changes into a state $s\prime$ according to the transition probability function $p(s\prime|s, a) \in \Delta(\mathcal{P})$. The initial state $s_0$ is sampled from the initial distribution $p_0(s_0)$ and $\gamma \in [0, 1)$ denotes the discount factor. The

objective of RL tasks is to learn a policy $\pi : \mathcal{S} \to \Delta(\mathcal{A})$ that maximize the expected discounted cumulative rewards (a.k.a return) along a trajectory:

$$\max_{\pi} \mathbb{E} \left[ \sum_{t=0}^{\infty} \gamma^t r(s_t, a_t \mid s_0 = s, a_0 = a) \right]$$

Value-based RL methods typically introduce a state-action value function, noted as $Q$-function, under approximate dynamic programming (Sutton et al., 1998; Haarnoja et al., 2018; Fujimoto et al., 2018). The temporal-difference (TD) learning is employed to learn the $Q$-function to satisfy the single-step Bellman consistency, minimizing the mean squared error between $Q_\pi(s, a)$ and its bootstrapped target $(\mathcal{T}^\pi)Q(s, a)$ with respect to the policy $\pi$:

$$(\mathcal{T}^\pi Q)(s, a) := r(s, a) + \gamma \mathbb{E}_{p(s'\mid s,a), \pi(a'\mid s')} \left[ Q_\pi(s', a') \right]. \tag{1}$$

## 4 IMPLICIT SPARSITY IN DENSE TRAINING OF DRL

In this section, we explore the growing implicit sparsity during traditional dense network training in DRL, which coincides with diminished plasticity in the networks. To measure this escalating sparsity (or reduced plasticity), we introduce the Weight Shrinkage Ratio (WSR) (Section 4.1). We depict the evolution of implicit sparsity (or reduced plasticity) in the network throughout training and advocate for the adoption of the dense-to-sparse training paradigm in the consideration of neural network plasticity (Section 4.2).

### 4.1 WEIGHT SHRINKAGE RATIO

Consider a deep neural network, denoted by $\mathbf{M}$, composed of $L$ hidden layers, where each layer is indexed by $l \in \{1, 2, \cdots, L\}$. Let us define $h^l$ as the weight vector from layer $l$ in the network $\mathbf{M}$, given an input dataset distribution $\mathcal{D}$. The number of neurons in each layer is represented by $N^l$. To explore the plasticity of neural networks, we propose a novel statistical metric:

**Definition 4.1** (Weight Shrinkage Ratio). For a given input distribution $\mathcal{D}$, the Weight Shrinkage Ratio (WSR) for layer $h_t^l$ is defined as the proportion of weights in $h_t^l$ that have decreased in magnitude from the current training step $t$ to its previous checkpoint step $t - k$ with $k \in (0, t)$:

$$\mathbb{D}[h_t^l \mid h_{t-k}^l] := \mathbb{E}_{x \in \mathcal{D}} \left[ \frac{\sum_{i=1}^{N^l} \mathbb{I}(|h_{t,i}^l(x)| < |h_{t-k,i}^l(x)|)}{N^l} \right], \tag{2}$$

where $\mathbb{I}(.)$ denotes the indicator function, returning 1 if the enclosed condition is true and 0 otherwise, and $i$ denotes the weight of the $i^{th}$ neuron.

The WSR for a model, such as a multi-layer neural network, can be computed through a straightforward summation across all hidden layers: $\mathbb{D}[\mathbf{M}_t \mid \mathbf{M}_{t-k}] := \sum_{i=l} \mathbb{D}[h_t^i \mid h_{t-k}^i]$. The purpose of WSR is to quantify the ratio of weights that have a lower magnitude at the current time step $t$ with respect to the last checkpoint step $t - k$. The normalization term in the denominator, $N^l$, ensures that the WSR is a dimensionless quantity. This normalization facilitates the comparison of WSR across different layers or networks by scaling the WSR accordingly. To illustrate the quantitative interpretation of WSR and the factors that contribute to it, we provide an intuitive example starting with Gaussian distribution, a commonly used distribution for initializing neural networks.

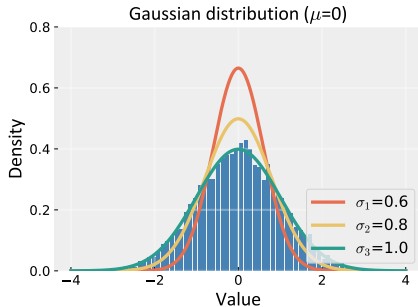

Figure 1: Gaussian distribution with the standard mean ($\mu = 0$) but different variances.

In Fig. 1, we examine three Gaussian distributions, each possessing an identical mean ($\mu = 0$) but differing in standard deviations. We specifically consider $(N_1, N_2, N_3) = (N(0, 0.6^2), N(0, 0.8^2), N(0, 1^2))$. Upon sampling 1000 data points from those distributions, we

yield $\mathbb{D}[N_2|N_1] = 56.8\%$ and $\mathbb{D}[N_3|N_2] = 59.4\%$. Note that $\mathbb{D}$ serves as an approximation of the shrinkage speed, scaled by a factor $k$, implying fractional shrinkage is initiated whenever $\mathbb{D} > 0$. The increased shrinkage speed from $N_2 \to D_3$ ($\mathbb{D}[N_3|N_2] = 59.4\%$) to $N_1 \to N_2$ ($\mathbb{D}[N_2|N_1] = 56.8\%$) indicates an acceleration in the convergence speed of data points towards zero.

## 4.2 IMPLICIT SPARSITY IN DENSE TRAINING

This section focuses first on demonstrating the generality of implicit sparsity in conventional dense training (fully connected neural networks) in DRL. Initially, we monitor the WSR throughout the dense training for two distinct task types. The first is high-dimension pixelated tasks with discrete action spaces, for which we employ DQN (Mnih et al., 2015) on the Atari platform (Mnih et al., 2013). The second is dynamic-based observation tasks with continuous action spaces, for which we utilize SAC (Haarnoja et al., 2018) on MuJoCo locomotion tasks (Todorov et al., 2012). All results in the following are averaged over 5 independent seeds with the standard deviation. For clarity, we simplify our analysis by setting $k$ equal to the evaluation frequency in different tasks: $5e3$ steps for MuJoCo locomotion tasks and $2e5$ steps for Atari games.

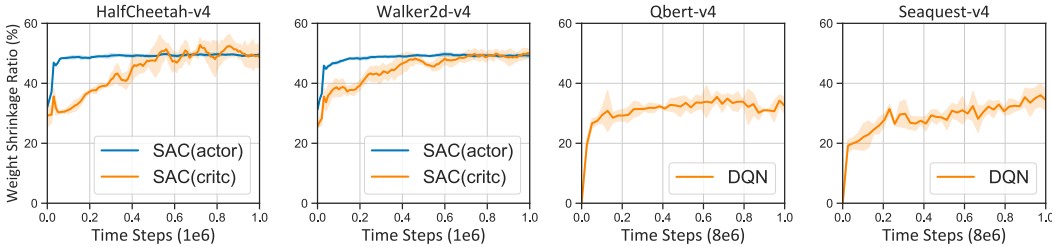

Figure 2: The WSR exhibits a growth pattern for both SAC and DQN networks throughout training.

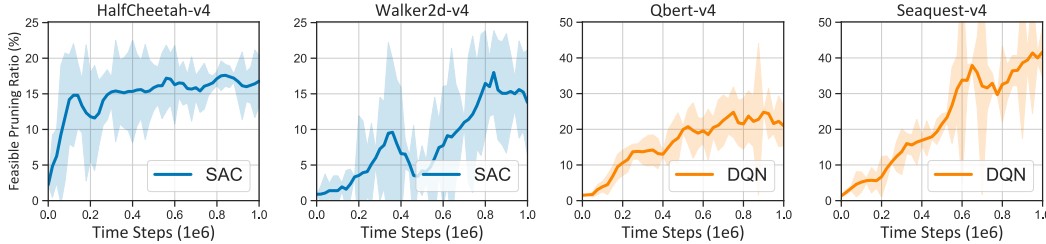

Figure 3: The feasible pruning ratio, indicative of the maximum pruning rate that allows models to retain at least $95\%$ performance relative to the dense model, increases throughout the training of both SAC and DQN agents.

**Increased implicit sparsity with training steps.** In Fig. 2, we discern a clear upward trajectory in WSR throughout the training process. The escalating pattern of WSR throughout training suggests an increasing speed of partial neural weight shrinkage towards 0 as training progresses. The rising overlap coefficient of the shrinkage weights, compared with the last checkpoint, suggests that this shrinkage trend of most weights persists throughout the remainder of the training process. Those trend remains consistent across a variety of algorithms and tasks, shown in Appendix A.

**Shrinkage persists across various activation functions.** One might hypothesize that the Rectified Linear Unit (ReLU) activation function (Nair & Hinton, 2010), which sets a lower bound of zero for negative input, contributes to this phenomenon. To probe this further, we first calculate the WSR using different activation functions that either lack a lower bound or have a negative lower bound, such as Leaky ReLU (Maas et al., 2013) and Sigmoid, respectively. We extend our investigation to the gradient shrinkage ratio by substituting the weight gradient for weight in Eqn. (2) under varying activation functions. Due to space constraints, we present these results in Appendix A. Our findings indicate that shrinkage occurs for both gradient and weight in neural networks across different activation functions. The SAC agent exhibits a consistent pattern across all activation functions. Specifically, the gradient shrinkage ratio rapidly escalates to nearly $50\%$ during the initial training

stage, and subsequently oscillates around this value for the remainder of the training period. This suggests that gradient shrinkage persists, albeit at a consistent rate.

**Increasing sparsity as training progresses.** However, it is crucial to understand that these diminishing neural weights or gradients may still contribute to the final representation and do not necessarily indicate a clear pattern of sparsity within neural networks. To delve deeper into this, we perform explicit neuron pruning to determine the feasible pruning rate. This rate represents the maximum pruning rate that allows models to maintain at least $95\%$ performance relative to the dense model. As illustrated in Fig. 3, we show the feasible pruning ratio increases in tandem with the progression of training steps, aligning significantly with the WSR trend. The consistency of the feasible pruning rate across various tasks is further elaborated in Appendix A.

The study by Sokar et al. (2023) presents a compelling finding: reinitializing weights (under the ReLU activate function) that approach zero beneath a specified threshold can significantly enhance performance over the course of training. The potential for improvement arises from addressing inactive or dormant neurons, which signifies a decrease in neural plasticity. In the realm of sparse training, the sparse-to-sparse training paradigm presents a trade-off: while it reduces computational memory demands at the initial training stages, it does so at the cost of the expressivity of neural networks. As a result, it could lead to the loss of plasticity of neural networks, especially at high sparsity ratios, even when subjected to dynamic changes. To address this, we propose a dense-to-sparse training paradigm that also enhances network plasticity at the very beginning, thereby improving the final performance even under high pruning ratios.

## 5 PLASTICITY-DRIVEN SPARSITY TRAINING

In the preceding section, we highlight an increase in implicit sparsity and a concurrent loss of plasticity during sparse DRL training. These observations motivate us to propose a new framework, Plasticity-Driven Sparsity Training (PlaD). PlaD adopts a dense-to-sparse training paradigm with the goal of enhancing performance in sparse DRL models by preserving neural plasticity throughout the training process. More specifically, PlaD is characterized by two key components: 1) periodic memory reset, which ensures consistency in the replay buffer and thereby improves the plasticity of DRL agents, and 2) dynamic weight rescaling (DWR), which is designed to counterbalance the instability introduced by the resetting and pruning operations.

**Periodic Memory Reset.** A naïve approach to maintaining plasticity throughout the training process in DRL involves periodic re-initialization of multiple complete neural networks of the agents while maintaining the experience within the buffer (Nikishin et al., 2022). However, this approach is notoriously resource-hungry due to the numerous re-initialization operations and a significantly high replay ratio, which is defined as the number of updates to parameters per environment interaction. Further, the high replay ratio paradoxically accelerates the loss of plasticity, leading to suboptimal performance. Other similar methods typically impose constraints on the neural networks, but these methods inevitably hamper the flow of gradients essential for policy updates.

Instead of directly modifying neural networks, we periodically reset the replay buffer to empty (0.2M) and then collect a batch of samples necessary for training, with the spirit of preserving the simplicity of our proposed algorithm. This strategy does not impact the policy gradient but effectively addresses non-stationarity, an important factor contributing to plasticity loss in DRL training (Sokar et al., 2023; Lyle et al., 2023), thereby maintaining policy consistency within the replay buffer. In the Appendix B.2, we illustrate that a straightforward memory reset effectively reduces the policy distance between the replay buffer and the current policy. Importantly, this operation does not impose an extra computational burden, such as determining the policy distance of the reply buffer at every training step (Tan et al., 2022).

**Dynamic Weight Rescaling (DWR).** In practice, the periodic memory reset, as well as the sparse training, impose the training instability over the course of training. Based on this motivation, we further introduce a supplement but necessary component in PlaD, namely dynamic weight recaling. Specifically, consider a sparse neural network $\mathbf{M_s}$, denoted as $\mathbf{M_s} = \{\Gamma^l : l = 1, \ldots, L\}$, which mirrors the structure of $\mathbf{M}$ in terms of weights, where $\gamma^l$ represents the mask applied to the $l^{th}$ layer. Consequently, the sparse network $\mathbf{M_s}$ can be represented as follows:

$$a^l = h^l \odot \gamma^l \quad u^{l+1} = f_l \left( a^{l^\top} u^l + b^l \right), \tag{3}$$

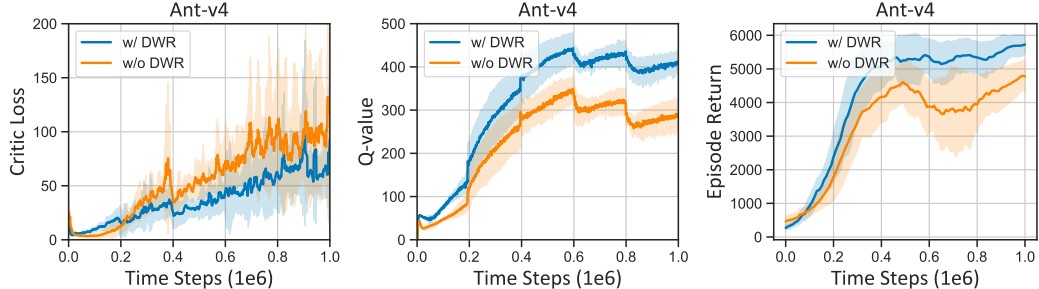

Figure 4: DWR mitigates the learning instability induced by memory reset and dynamic training. **Left:** PlaD (w/o DWR) typically exhibits higher instability, as evidenced by increased critic loss and variances in Bellman updates. **Middle:** The Q-value of PlaD (w/ DWR) is rationally higher than PlaD (w/o DWR), potentially leading to improved policy performance. **Right**: The performance PlaD (w/o DWR) significantly falls short when compared to the performance with PlaD (w/ DWR).

where $a^l$ is the pruned (or masked) neuron weights, $\odot$ is the element-wise product, $u^l$ represents the input vector to the $l$-th layer, $b^l$ is the bias, and $f_l$ is the transformation function for the $l^{th}$ layer. After getting pruned weights, We can readily obtain the dynamic statistical information, namely, the mean and standard variance across all hidden units within the same layers:

$$\mu^l = \frac{1}{L} \sum_{i=1}^{L} a_i^l \quad \sigma^l = \sqrt{\frac{1}{L} \sum_{i=1}^{L} \left(a_i^l - \mu^l\right)^2}.$$

We then dynamic scale weights that are not been pruned:

$$\hat{a}^l = \frac{a^l - \mu^l}{\sqrt{\left(\sigma^l\right)^2 + \epsilon}}, \tag{4}$$

where $\epsilon$ is a small number of significance introduced to prevent the denominator from becoming zero. Dynamic Weight Rescaling (DWR) exhibits properties akin to those of layer normalization (Ba et al., 2016); however, a notable distinction lies in their operational domains. While DWR applies to pruned weights $a^l$ during sparse training, layer normalization functions on $(a^l)^T u^l$. As depicted in Fig. 4, DWR mitigates the learning instability caused by memory reset and sparse training. Consistent observations across different tasks can be found in the ablation study in Section 6.2. We observe that the critic loss of PlaD (w/ DWR) is consistently lower than the critic loss PlaD (w/o DWR) as training progresses. The occurrence of lower critic loss but higher Q-value in PlaD (with DWR) suggests that the higher Q-value effectively enhances the flow of gradients, thereby resulting in superior performance compared to PlaD (w/o DWR).

## 6 EXPERIMENTS

We conducted experiences to assess and analyze for PlaD. In Section 6.1, we first evaluate PlaD on standard MuJoCo environments with other sparse training baselines. Section 6.2 contains an ablation study demonstrating the necessity of both components in PlaD for policy improvement. Lastly, in Section 6.3, we analyze the effect of buffer size, comparing a periodic memory reset in PlaD to a smaller buffer without the memory reset. More details in experiments are shown in Appendix B.

### 6.1 PERFORMANCE ON BENCHMARKS

We perform a standard benchmark comparison of PlaD with a range of other sparse training methods. This comparison, which is conducted within the context of MuJoCo environments using the Soft Actor-Critic (SAC) as a backbone, is detailed in Fig. 5. The comparative baselines encompass a diverse set of sparse training techniques, including both dense-to-sparse (solid lines) and sparse-to-sparse training paradigms (dotted lines). The dense-to-sparse baselines all initialize with a dense network, including: (1) **Random**: the most naïve baseline to randomly iterative pruning the weights. (2) **Magnitude** (Frankle & Carbin, 2019): performing iterative weight pruning as the training goes. On the other hand, the sparse-to-sparse training paradigm initializes a sparse network to the target

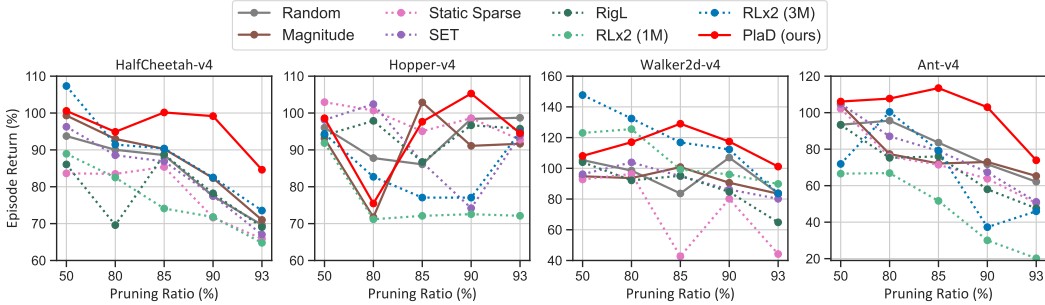

Figure 5: Performance comparisons of PlaD with sparse training baselines with the SAC backbone, normalized with the performance achieved by vanilla SAC, where the solid line and dotted line indicate the dense-to-sparse and sparse-to-sparse training paradigm, respectively. PlaD achieves the best performance in 10 out of 12 tasks with high pruning ratios ($\geq 85\%$) in different environments.

sparsity ratio before training, including: (1) **Static Sparse** (Lee et al., 2019): pruning a given dense network randomly at initialization and the resulting sparse network is trained with a fixed structure. (2) **SET** (Mocanu et al., 2018): Using the dynamic sparse training, a portion of the connections are periodically changed by the replacement of connections characterized by the lowest magnitudes with new, randomly initialized ones. (3) **RigL** (Evci et al., 2020): the same as SET, except the new connections are activated according to the highest magnitude of gradient signal instead of random. (4) **RLx2**: the same as RigL, except for two specific RL components for robust value learning to mitigate non-stationary, where the following content in the bracket refers to the number of training steps, such as 3M refers to 3 million training steps, while others are 1 million training steps otherwise specified. For the fairness of comparison, we specify the pruning ratio as the same for both actor and critic networks. For all algorithms under consideration, we employ the ERK network distribution (Evci et al., 2020) due to its superior efficiency compared to uniform distribution (Graesser et al., 2022). More experiment details in benchmark experiments are displayed in Appendix B and benchmark tables with the standard deviation are shown in Appendix B.3.

As evidenced in Fig. 5, our algorithm, PlaD, exhibits a significant performance superiority over other baselines. This superiority becomes more pronounced at high pruning ratios ($\geq 85\%$), where PlaD outperforms other baselines in 10 out of 12 tasks. For instance, in the `HalfCheetah` task at 90% sparsity, PlaD achieves a remarkable performance increase, outstripping the nearest baseline (RLx3 (3M)) by nearly 17%, reaching 99.2% compared to 82.5%. Similarly, in the `Ant` task with 90% sparsity, PlaD's performance of 103.0% surpasses the best baseline (Magnitude) by a substantial 30%, the latter achieving only 71.7%. The pronounced performance of PlaD relatively mediocre performance at lower pruning ratios such as $50\%$, can be attributed to the less apparent loss of plasticity at lower ratios. However, this plasticity loss becomes more conspicuous and impactful at higher pruning ratios, thus highlighting the strengths of PlaD.

Interestingly, we observe that PlaD achieves its peak performance within the high range of $85\%$ to $90\%$ pruning ratios. This performance not only matches but also surpasses that of the corresponding dense model derived from the SAC algorithm by a large margin. For instance, in the `Walker2d` task, PlaD achieves an impressive approximate $130\%$ of the performance of the dense model at an $85\%$ pruning ratio in the `Ant` task. Furthermore, our analysis reveals that the sparse-to-sparse training paradigm demands substantial computational resources to achieve performance levels comparable to those of the dense-to-sparse training paradigm. For example, while the performance of RLx2 (3M) is on par with other dense models, the performance of RLx2 (1M) is lower than the baselines derived from the dense-to-sparse paradigm in most tasks at different pruning ratios.

## 6.2 THE TWO COMPONENTS ARE NECESSARY

To underscore the critical roles of memory reset and DWR within PlaD, we conduct an ablation study at high pruning ratios, as shown in Tab. 1. The results show that the PlaD (w/o DWR) leads to diminished performance and increased variances in tasks such as `Hopper-v4` and `Ant-v4`. It underscores the importance of prioritizing training stability when sparse training is integrated with memory reset. Conversely, PlaD (w/o Reset) exhibits performance levels similar to the Magnitude method, but with reduced variances in most tasks. This outcome attests to the effectiveness of DWR in stabilizing the training process. Within this combined approach, memory reset plays a crucial role

in enhancing performance at high sparsity ratios by preserving model plasticity. Concurrently, DWR effectively mitigates the training instability from memory reset and sparse training, thereby bolstering the overall performance of PlaD.

Table 1: An ablation study on memory reset and subsequent DWR in PlaD, where performance (%) is normalized and compared to the performance from its corresponding dense model over 5 independent seeds, including standard deviation.

| Algorithms | Sparsity | HalfCheetah-v4 | Hopper-v4 | Walker2d-v4 | Ant-v4 |
|---|---|---|---|---|---|
| Magnitude | | 82.3±13.6 | 91.1±6.7 | 90.6±8.4 | 72.9±14.0 |
| PlaD (w/o DWR) | | 86.2±18.4 | 80.5±20.3 | 98.0±13.7 | 68.2±17.7 |
| PlaD (w/o Reset) | 0.9 | 85.4±5.8 | 92.3±3.1 | 96.5±4.5 | 77.0±6.3 |
| PlaD | | 99.2 ±3.9 | 105.3 ±7.3 | 117.4 ±5.2 | 103.5 ±6.5 |
| Magnitude | | 71.0±11.3 | 91.6±12.8 | 84.5±15.7 | 65.5±7.2 |
| PlaD (w/o DWR) | | 81.9±13.5 | 78.8±19.7 | 96.3±18.7 | 55.5±12.5 |
| PlaD (w/o Reset) | 0.93 | 73.5±9.2 | 92.9±4.4 | 83.6±8.2 | 71.6±15.3 |
| PlaD | | 84.6 ±7.6 | 94.5 ±12.5 | 106.7 ±11.4 | 78.4 ±7.5 |

## 6.3 CAN WE USE A SMALLER REPLY BUFFER?

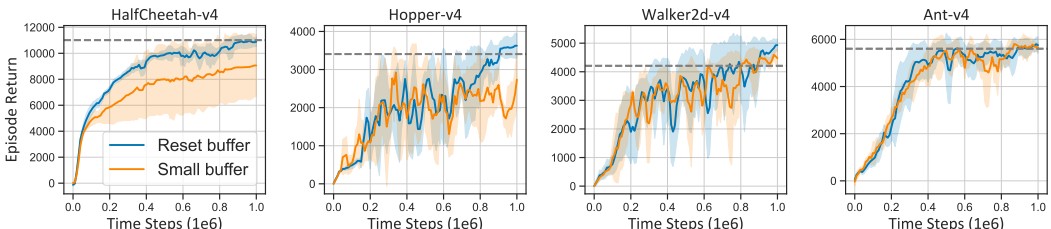

Figure 6: Performance comparison between `Reset buffer` and `Small buffer` in PlaD with 0.9 target sparsity. The results are averaged over five independent seeds, with the standard deviation indicated. Black dotted lines represent the dense performance obtained from the vanilla SAC algorithm. `Reset buffer` outperforms the `Small Buffer` strategy in 3 out of 4 tasks in terms of final averaged performance, with a large margin in 2 of them.

An intriguing aspect warranting further exploration pertains to PlaD is the operation of the replay buffer. Given that direct memory reset leads to significant challenges in training stabilization, one might consider employing a smaller buffer size as a potential solution. To investigate this, we compare these two settings (`Reset buffer` vs. `Small buffer`) in a high pruning ratio (90%) with 0.2M buffer size, as shown in Fig. 6. Our results indicate that `Reset buffer` significantly surpasses `Small Buffer` in 3 out of 4 tasks, most notably in the `Hopper` task over 30% gains averaged with dense performance. `Reset buffer` periodically imposes a steep learning curve on the agent, thereby facilitating the learning of relatively fresh experiences, compared with the gentle learning curve in `Small Buffer`. Such a dynamic learning curve approach in `Reset buffer` can be beneficial when the policy needs to undergo significant evolution during training, particularly in the context of non-stationary data flows. Consistent results with an extremely high sparsity ratio (93%) can be found in Appendix B.4.

## 7 CONCLUSIONS AND LIMITATIONS

In this study, we initially establish a link between the loss of plasticity and sparse training. Subsequently, we introduce a novel dense-to-sparse training algorithm for sparse training in DRL, referred to as PlaD, with the primary motivation to enhance network plasticity. PlaD employs memory reset to mitigate the non-stationarity in the replay buffer, which is a primary factor contributing to the loss of plasticity in DRL. Furthermore, PlaD introduces dynamic weight rescaling (DWR) to stabilize the training process, which could otherwise be disrupted by memory reset and sparse training. Our extensive evaluations show the state-of-the-art sparse training performance and highlight the essential

for those two components. Surprisingly, we find that PlaD is capable of achieving higher performance than the dense performance in high sparsity ratios due to the plasticity perspective. One limitation of PlaD is the lack of theoretical analysis and we hope this work will shed light on future rigorous analysis between sparse training and the loss of plasticity in DRL. We also hope this work could inspire more attention to real-world applications characterized by constrained resources or latency.

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

# A    IMPLICIT SPARSE IN DENSE TRAINING

This section provides a comprehensive elaboration related to Section 4. It encompasses additional consistent experiments that illustrate the increasing trend of the Weight Shrinkage Ratio (WSR) for both continuous and discrete action spaces throughout the training process. Furthermore, it includes ablation studies on the activation functions for both the weight and gradient in neural networks, as well as an examination of the feasible pruning ratio. All the reported performances are averaged over 5 independent seeds with the standard deviation.

**Weight Shrinkage Ratio (WSR).**    Fig. 7 and Fig. 8 illustrate the consistently increasing trend in both continuous and discrete environments. In the continuous task domains, a rapid increase in the critic network is evident compared to the actor network. It's noteworthy that shrinkage occurs when the WSR is greater than or equal to zero.

In order to delve deeper into the consistency of the "shrinkage" effect observed in certain neurons as training progresses, we monitor the overlap coefficient of these neurons compared to the final network checkpoint. The overlap coefficient[1] for both SAC and DQN agents is illustrated in Fig. 9 and Fig. 10. The observed increase strongly implies that once a neuron enters a state of shrinkage, it maintains this status throughout the remainder of the training process. Importantly, we note that the approximation of a $100\%$ overlap coefficient for the critic network strongly indicates that once neurons embark on the shrinkage process, they are highly likely to persist in this state for the duration of the remaining training steps.

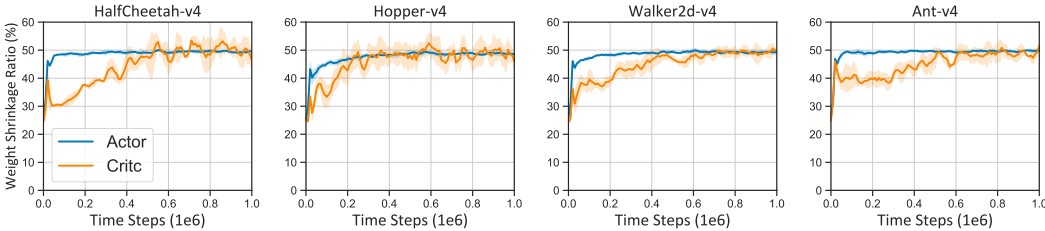

Figure 7: The WSR exhibits a growth pattern for SAC networks throughout training with the ReLU activate function.

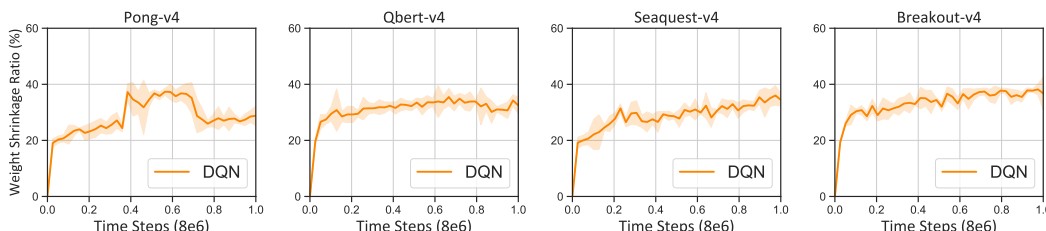

Figure 8: The WSR exhibits a growth pattern for DQN networks throughout training with the ReLU activate function.

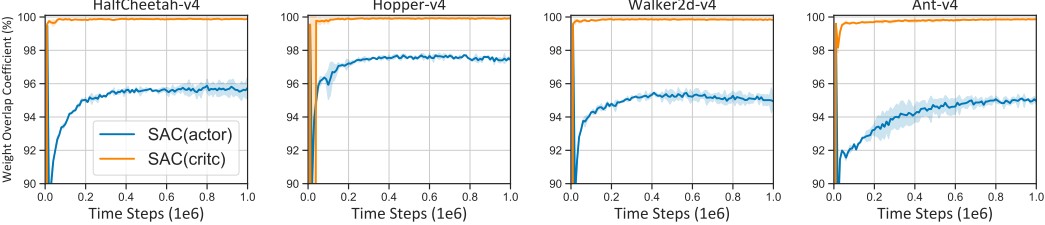

Figure 9: The weight overlap coefficient of SAC networks throughout training with the ReLU activate function.

**Ablation on Different Activation Functions.**    We first conduct an ablation study on the activation function with respect to the WSR. Fig. 11, Fig. 12, Fig. 13, and Fig. 14 illustrate a consistent

---

[1]The overlap coefficient between two sets X and Y is defined as $\mathrm{overlap}(X, Y) = \frac{|X \cap Y|}{\min(|X|, |Y|)}$

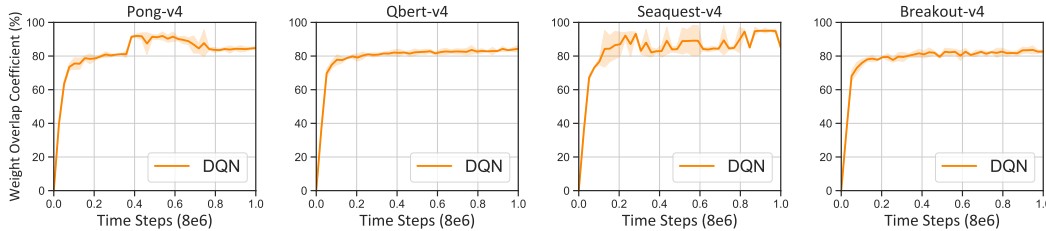

Figure 10: The weight overlap coefficient of DQN networks throughout training with the ReLU activate function.

increasing pattern for WSR, indicating that shrinkage occurs regardless of the activation function used in continuous and discrete tasks. Interestingly, we observe a pulse occurring in most tasks within DQN when employing the Sigmoid activation function, except the `Seaquest` task. This suggests a rapid increase in weight shrinkage during the initial learning period, as shown in Fig. 14.

We further validate this with the gradient shrinkage ratio, for both continuous and discrete tasks, where we observe a significant alignment between the weight shrinkage ratio and gradient shrinkage ratio across different tasks and activation functions (as shown in Fig. 16, Fig. 17, Fig. 18, Fig. 19, and Fig. 20). For the gradient shrinkage ratio, both DQN and SAC agents exhibit a stable degree of shrinkage. For example, the gradient shrinkage ratio in MuJoCo tasks swiftly rises to nearly $50\%$ during the initial stages of training, and then fluctuates around $50\%$ for the remainder of the training period. This indicates that gradient shrinkage continues to take place at a steady pace.

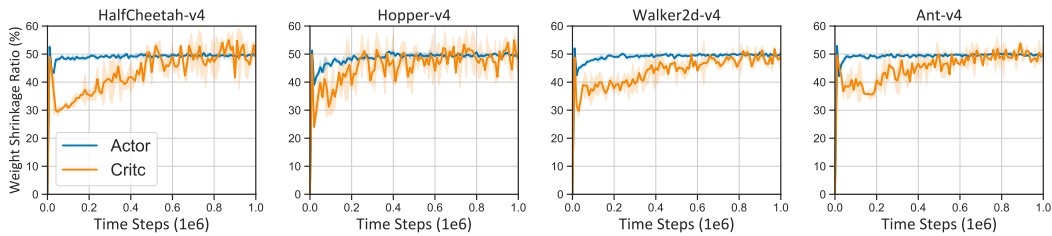

Figure 11: The WSR exhibits a growth pattern for SAC networks on the Leaky ReLU activate function throughout training.

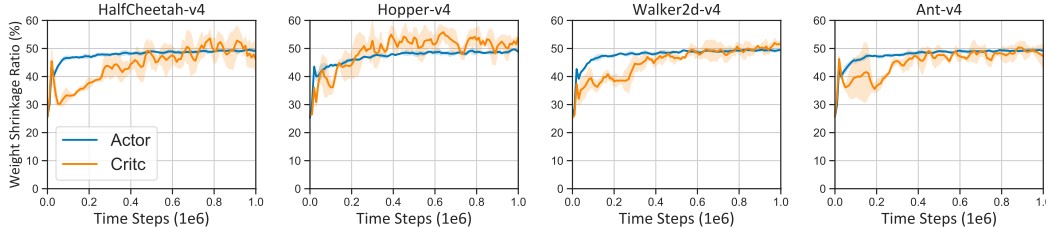

Figure 12: The WSR exhibits a growth pattern for SAC networks on the Sigmoid activate function throughout training.

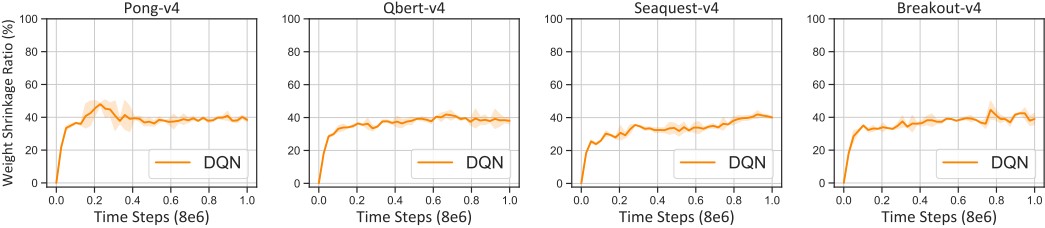

Figure 13: The WSR exhibits a growth pattern for DQN networks on the Leaky ReLU activate function throughout training.

It's important to note that the "increasing" pattern may occur at different stages due to the influence of tasks or algorithms. For instance, the `HalfCheetah-4` task displays this during the initial training

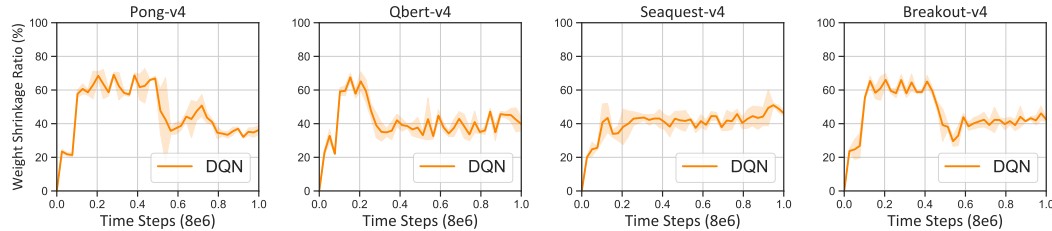

Figure 14: The WSR exhibits a growth pattern for DQN networks on the Sigmoid activate function throughout training.

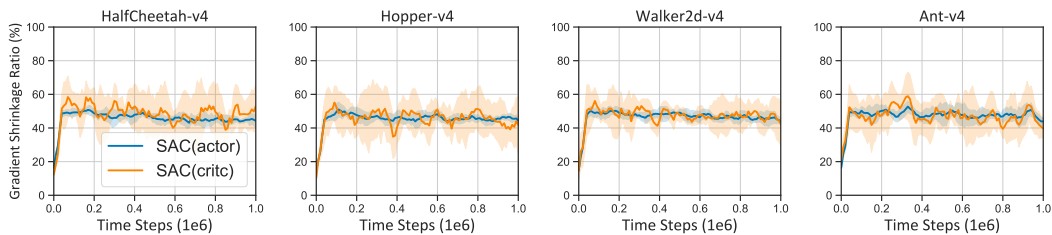

Figure 15: The gradient shrinkage ratio exhibits a growth pattern for SAC networks on the ReLU activate function throughout training.

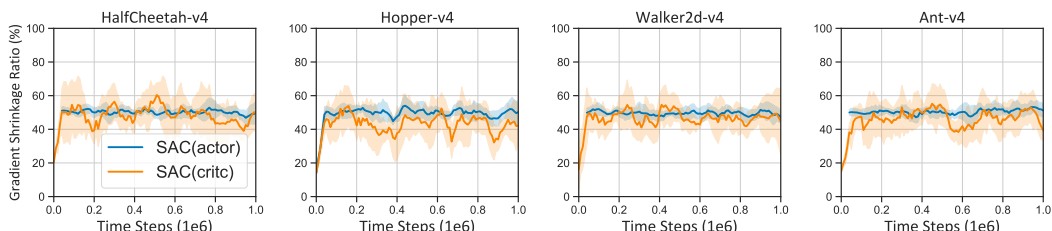

Figure 16: The gradient shrinkage ratio exhibits a growth pattern for SAC networks on the Leaky ReLU activate function throughout training.

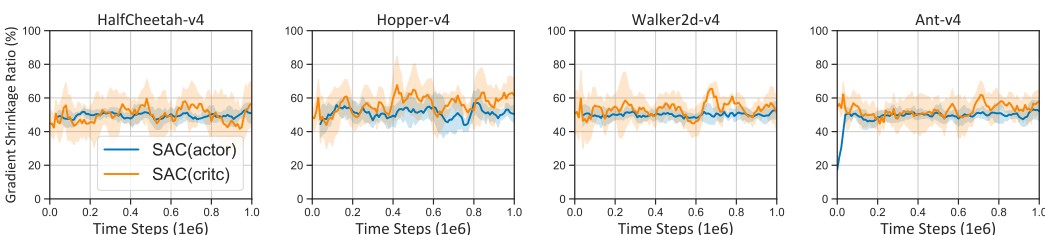

Figure 17: The gradient shrinkage ratio exhibits a growth pattern for SAC networks on the Sigmoid activate function throughout training.

stage and the `Walker-2d` task throughout the entire training process. Nevertheless, a clear increase in the feasible pruning ratio over the course of training is observable in all cases.

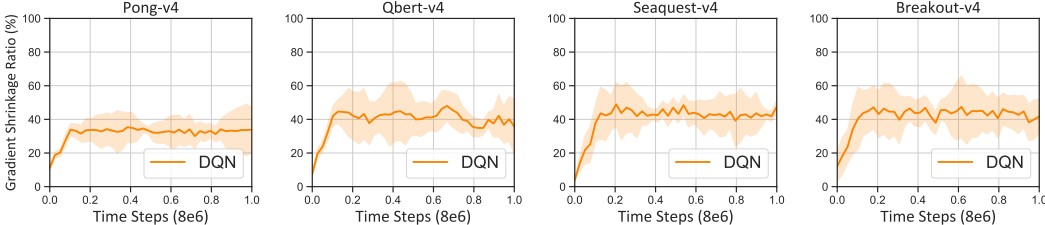

Figure 18: The gradient shrinkage ratio exhibits a growth pattern for DQN networks on the ReLU activate function throughout training.

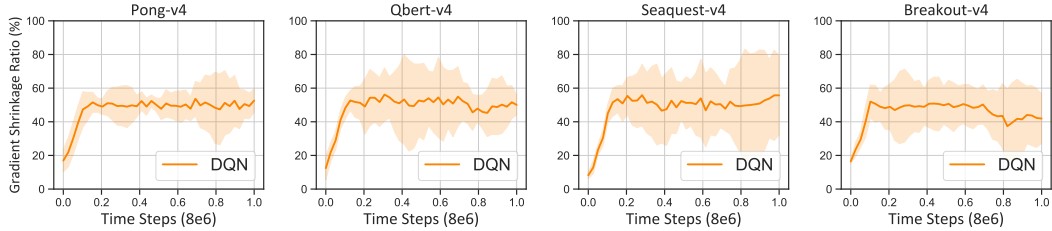

Figure 19: The gradient shrinkage ratio exhibits a growth pattern for DQN networks on the Leaky ReLU activate function throughout training.

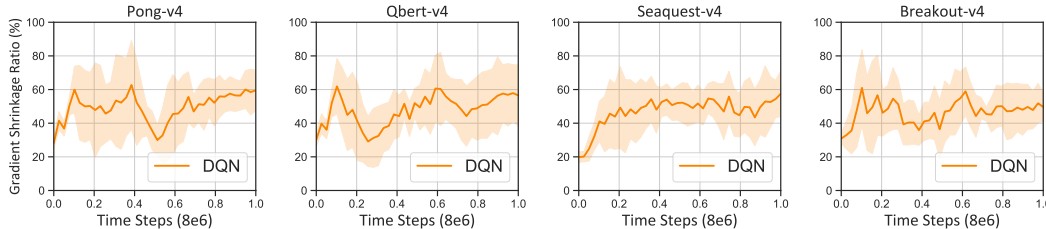

Figure 20: The gradient shrinkage ratio exhibits a growth pattern for DQN networks on the Sigmoid activate function throughout training.

**Feasible Pruning Ratio (%).**     We provide comprehensive experiments on feasible pruning across both discrete (DQN Atari) and continuous environments (Gym MuJoCo and Deepmind Control Suites) in Fig. 21, Fig. 22, and Fig. 23.

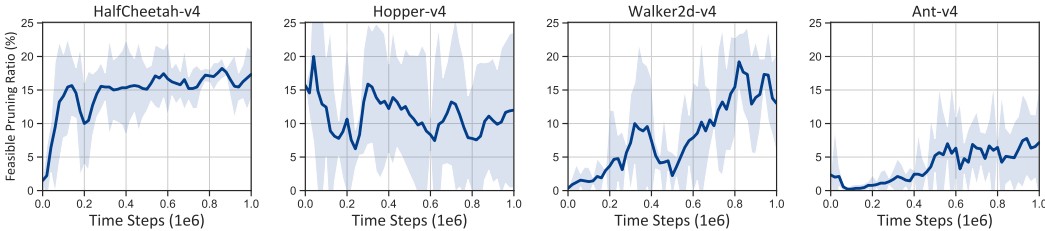

Figure 21: The feasible pruning ratio increases throughout the training for SAC agents in MuJoCo locomotion tasks.

## B    EXPERIMENTS DETAILS

This section offers additional experimental details related to Section 6, including a comprehensive experimental setup, network structure, and an ablation study on memory reset with an extremely high sparsity ratio (0.93).

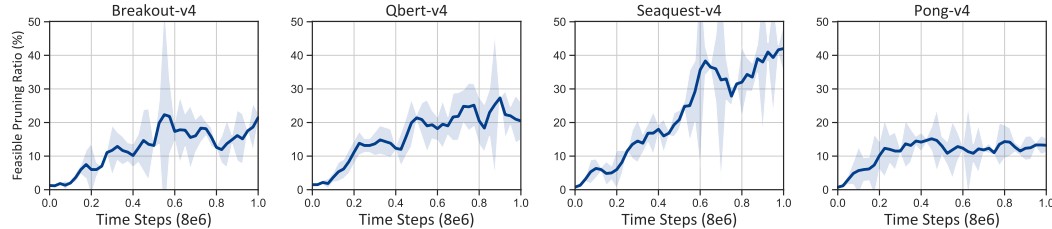

Figure 22: The feasible pruning ratio increases throughout the training for DQN agents in Atari games.

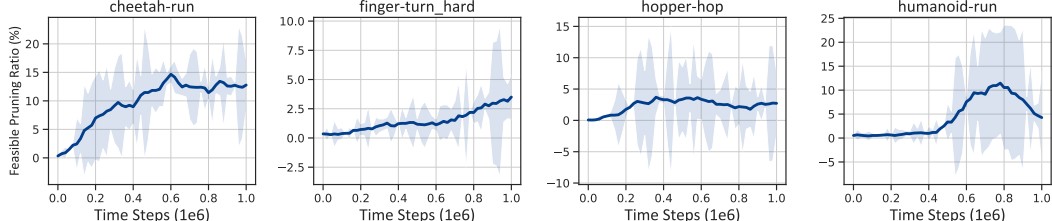

Figure 23: The feasible pruning ratio increases throughout the training for SAC agents in Deepmind control suites.

## B.1 HARDWARE SETUP

Our experiments were conducted using JAX (Bradbury et al., 2018) and executed on 4x NVIDIA GeForce RTX 3090. The implementations of Jax DQN and SAC were based on CleanRL (Huang et al., 2022) and the open-source Jax SAC (Kostrikov, 2021) implementation, respectively. The pruning algorithms were based on Jaxpruner (Lee et al., 2023), with the exception of RLx2, which was based on the author's official implementation. All training processes involved 1 million steps, except for RLx2 (3M), which entailed 3 million steps.

## B.2 PERIODIC MEMORY RESET ENSURES THE POLICY CONSISTENCE

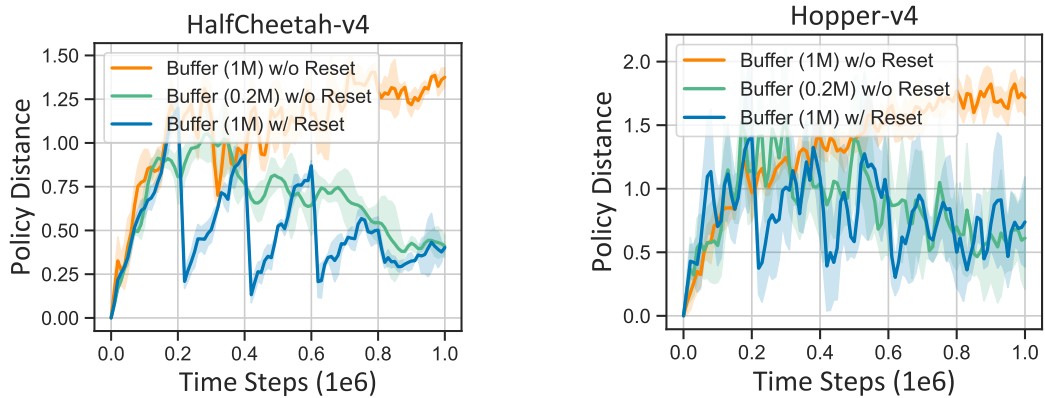

Figure 24: The policy distance between the reply buffer and the current learning policy.

In this section, we present evidence demonstrating the impact of periodic memory resets on maintaining policy consistency between the learned policy and the behavior policy within the replay buffer. Expanding on the methodology of (Tan et al., 2022), we employ a policy distance metric to measure the disparity between the data in the buffer and the current learning policy:

$$\mathcal{D}(\mathcal{B}, \phi) = \frac{1}{K} \sum_{(s_i, a_i) \in \mathrm{OldK}(\mathcal{B})} \left\| \pi\left(s_i; \phi\right) - a_i \right\|_2,$$

Here, $\mathcal{B}$ signifies the replay buffer, $\text{OldK}(\mathcal{B})$ denotes the oldest $K$ transitions in $\mathcal{B}$, and $\pi(\cdot; \phi)$ represents the current policy. The hyperparameter $K$ is established as 256, corresponding to the batch size, in our experiments. A visualization of policy distance throughout the training process in Fig. 24 reveals that resets can effectively reduce the discrepancy between the current policy and the oldest $K$ transitions in the replay buffer. We also observe that the discrepancy can be somewhat alleviated by employing a smaller buffer. Note that this metric approximates the policy distance between the replay buffer and the current policy under the monotonic improvement assumption with a strong algorithm, where the oldest transitions are assumed to exhibit the maximum discrepancy between the behavior policy and the current policy.

## B.3 BENCHMARK TABLES

Table 2: Performance comparisons on `HalfCheetah` environment of PlaD with sparse training baselines with the SAC backbone, normalized with the performance achieved by vanilla SAC. All results are averaged over 5 independent seeds and reported with the standard deviation. PlaD achieves the best performance in 4 out of 5 pruning ratios with a significant margin, over 10%, in high sparsity ratios (85%, 90%, 93%) sparsity ratio over the best performance from baselines.

| Sparsity | 50% | 80% | 85% | 90% | 93% |
|---|---|---|---|---|---|
| Random | $93.8 \pm 6.0$ | $90.0 \pm 3.0$ | $88.6 \pm 1.0$ | $77.5 \pm 4.7$ | $69.7 \pm 0.2$ |
| Magnitude | $99.3 \pm 7.9$ | $93.0 \pm 6.2$ | $90.3 \pm 3.5$ | $82.3 \pm 13.6$ | $71.0 \pm 11.4$ |
| Static Sparse | $83.6 \pm 8.1$ | $83.5 \pm 9.6$ | $85.4 \pm 6.9$ | $71.6 \pm 3.2$ | $65.9 \pm 2.9$ |
| SET | $96.3 \pm 6.9$ | $88.6 \pm 6.8$ | $86.9 \pm 4.5$ | $77.6 \pm 2.7$ | $67.0 \pm 4.1$ |
| RigL | $86.1 \pm 5.0$ | $69.6 \pm 4.3$ | $88.8 \pm 3.5$ | $78.2 \pm 4.4$ | $69.0 \pm 3.5$ |
| RLx2 (1M) | $88.9 \pm 3.2$ | $82.5 \pm 3.5$ | $74.1 \pm 11.6$ | $71.8 \pm 5.1$ | $64.8 \pm 2.5$ |
| RLx2 (3M) | $107.3 \pm 3.8$ | $91.5 \pm 5.0$ | $90.4 \pm 1.7$ | $82.5 \pm 4.4$ | $73.5 \pm 12.2$ |
| PlaD (ours) | $100.6 \pm 6.9$ | $94.9 \pm 8.5$ | $100.2 \pm 5.5$ | $99.2 \pm 4.0$ | $\mathbf{84.6} \pm 7.6$ |

Table 3: Performance comparisons on the `Hopper` environment of PlaD with sparse training baselines with the SAC backbone, normalized with the performance achieved by vanilla SAC. All results are averaged over 5 independent seeds and reported with the standard deviation.

| Sparsity | 50% | 80% | 85% | 90% | 93% |
|---|---|---|---|---|---|
| Random | $96.0 \pm 12.7$ | $87.8 \pm 25.8$ | $86.1 \pm 27.5$ | $98.4 \pm 1.4$ | $98.7 \pm 5.4$ |
| Magnitude | $93.1 \pm 8.6$ | $71.7 \pm 42.5$ | $102.9 \pm 1.5$ | $91.1 \pm 6.7$ | $91.7 \pm 2.9$ |
| Static Sparse | $103.0 \pm 19.2$ | $100.7 \pm 3.0$ | $95.0 \pm 1.6$ | $98.6 \pm 13.9$ | $92.8 \pm 1.8$ |
| SET | $98.2 \pm 22.1$ | $102.4 \pm 3.0$ | $86.6 \pm 27.9$ | $74.2 \pm 34.0$ | $93.9 \pm 2.4$ |
| RigL | $94.0 \pm 19.6$ | $97.9 \pm 4.6$ | $86.7 \pm 23.7$ | $96.7 \pm 5.5$ | $95.8 \pm 2.7$ |
| RLx2 (1M) | $91.8 \pm 6.4$ | $71.2 \pm 20.4$ | $72.1 \pm 21.8$ | $72.6 \pm 28.4$ | $72.1 \pm 11.7$ |
| RLx2 (3M) | $94.4 \pm 11.3$ | $82.7 \pm 21.9$ | $77.1 \pm 29.0$ | $77.0 \pm 12.6$ | $94.9 \pm 1.5$ |
| PlaD (ours) | $98.6 \pm 9.4$ | $75.5 \pm 13.3$ | $97.6 \pm 4.8$ | $105.3 \pm 7.2$ | $94.5 \pm 9.5$ |

## B.4 CAN WE USE A SMALLER REPLY BUFFER?

We present the results for an extreme target sparsity of 0.93 in comparison with `Reset buffer` and `Small buffer`, as shown in Fig. 25. The outcomes align closely with the scenarios involving a target sparsity of 0.9 as shown in Fig. 6. Even with a target sparsity of 0.93, we can still observe a significant advantage of the `Reset buffer` strategy in the `HalfCheetah` and `Hopper` tasks.

Table 4: Performance comparisons on the `Walker2d` environment of PlaD with sparse training baselines with the SAC backbone, normalized with the performance achieved by vanilla SAC. All results are averaged over 5 independent seeds and reported with the standard deviation. It notably excels in 4 out of 5 pruning ratios, particularly distinguishing itself in the 85% and 93% sparsity domains with a near 10% performance lead over the best performance in baselines.

| Sparsity | 50% | 80% | 85% | 90% | 93% |
|---|---|---|---|---|---|
| Random | $105.4 \pm 9.2$ | $98.5 \pm 10.5$ | $83.6 \pm 3.4$ | $107.0 \pm 6.3$ | $83.8 \pm 3.0$ |
| Magnitude | $94.8 \pm 8.2$ | $93.6 \pm 13.1$ | $100.8 \pm 12.2$ | $90.6 \pm 8.5$ | $83.5 \pm 15.9$ |
| Static Sparse | $92.8 \pm 15.5$ | $96.3 \pm 7.1$ | $42.7 \pm 38.7$ | $80.2 \pm 9.0$ | $44.2 \pm 16.7$ |
| SET | $96.2 \pm 7.9$ | $103.9 \pm 22.9$ | $94.9 \pm 4.0$ | $86.2 \pm 12.5$ | $80.2 \pm 20.8$ |
| RigL | $104.0 \pm 12.7$ | $92.2 \pm 7.7$ | $95.2 \pm 11.2$ | $84.7 \pm 13.4$ | $64.8 \pm 19.3$ |
| RLx2 (1M) | $123.0 \pm 11.8$ | $125.5 \pm 8.0$ | $99.2 \pm 9.4$ | $96.0 \pm 11.2$ | $89.8 \pm 20.0$ |
| RLx2 (3M) | $147.7 \pm 8.9$ | $132.4 \pm 11.1$ | $116.9 \pm 12.4$ | $112.3 \pm 12.5$ | $83.6 \pm 13.1$ |
| PlaD (ours) | $108.1 \pm 11.7$ | $117.0 \pm 20.1$ | $129.0 \pm 11.8$ | $117.4 \pm 5.1$ | $101.1 \pm 11.4$ |

Table 5: Comparative performance analysis in the `Ant` environment, showcasing PlaD's superior efficiency over other sparse training methods with the SAC backbone. Results, normalized against vanilla SAC, are averaged over 5 seeds and include standard deviations. Notably, PlaD outperforms competitors across all pruning ratios, particularly excelling at higher sparsity levels (85% and 90%), where it achieves a performance increase of nearly 20%.

| Sparsity | 50% | 80% | 85% | 90% | 93% |
|---|---|---|---|---|---|
| Random | $93.4 \pm 10.3$ | $95.6 \pm 7.9$ | $83.6 \pm 7.6$ | $71.7 \pm 8.0$ | $62.2 \pm 14.9$ |
| Magnitude | $104.4 \pm 6.0$ | $77.3 \pm 5.8$ | $72.1 \pm 4.7$ | $73.0 \pm 14.2$ | $65.3 \pm 7.3$ |
| Static Sparse | $102.2 \pm 12.7$ | $75.9 \pm 11.3$ | $71.4 \pm 27.1$ | $63.8 \pm 19.6$ | $50.4 \pm 14.1$ |
| SET | $104.9 \pm 6.5$ | $87.0 \pm 9.9$ | $79.0 \pm 20.0$ | $67.4 \pm 23.2$ | $51.1 \pm 5.3$ |
| RigL | $93.3 \pm 11.8$ | $75.2 \pm 15.4$ | $76.0 \pm 16.0$ | $58.0 \pm 9.8$ | $47.6 \pm 13.3$ |
| RLx2 (1M) | $66.6 \pm 16.3$ | $66.9 \pm 18.2$ | $51.7 \pm 12.1$ | $30.0 \pm 11.2$ | $20.2 \pm 3.2$ |
| RLx2 (3M) | $71.9 \pm 10.1$ | $100.4 \pm 8.0$ | $79.1 \pm 23.5$ | $37.3 \pm 19.0$ | $46.0 \pm 6.9$ |
| PlaD (ours) | $106.1 \pm 8.0$ | $107.7 \pm 6.1$ | $113.5 \pm 3.7$ | $103.0 \pm 6.6$ | $73.9 \pm 7.5$ |

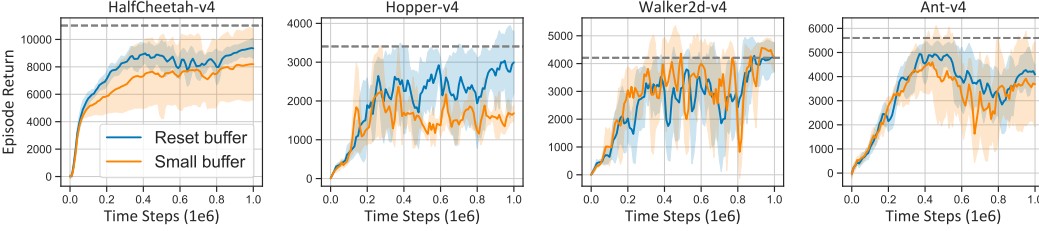

Figure 25: Performance comparison between `Reset buffer` and `Small buffer` in PlaD with 0.93 target sparsity.

## B.5 MODEL DETAILS FOR DQN, SAC, AND PLAD.

The parameter details of DQN and SAC agents, as well as PlaD, are shown in Tab. 6 and Tab. 7.

Table 6: DQN Hyperparameters.

| Parameter | Value |
|---|---|
| *DQN* | |
| optimizer | Adam (Kingma & Ba, 2015) |
| learning rate | $10^{-4}$ |
| discount ($\gamma$) | 0.99 |
| replay buffer size | $10^6$ |
| number of hidden layers (all networks) | 3 |
| number of hidden (all networks) | [64, 64, 512] |
| number of samples per minibatch | 32 |
| train frequency | 4 |
| target update interval | 1000 |
| exploration fraction | 0.1 |
| learning starts | 80000 |
| nonlinearity | ReLU |
| target update interval | 1000 |
| gradient steps | 1 |
| total learning steps | 1000000 |

Table 7: SAC and PlaD Hyperparameters.

| Parameter | Value |
|---|---|
| *SAC* | |
| optimizer | Adam |
| learning rate | $3 \cdot 10^{-4}$ |
| discount ($\gamma$) | 0.99 |
| replay buffer size | $10^6$ |
| number of hidden layers (all networks) | 2 |
| hidden units (all networks) | [256, 256] |
| number of samples per minibatch | 256 |
| nonlinearity | ReLU |
| target smoothing coefficient ($\tau$) | 0.005 |
| target update interval | 1 |
| gradient steps | 1 |
| total learning steps | 8000000 |
| *PlaD* | |
| memory reset interval | 20000 |
| collection after memory reset | 5000 |

