# OpenReview forum: "Plasticity-Driven Sparsity Training for Deep Reinforcement Learning"
_ICLR.cc/2024/Conference — Submitted to ICLR 2024_

### Official Review · Reviewer_aeZw · 2023-10-28

**Soundness:** 2 fair
**Presentation:** 2 fair
**Contribution:** 2 fair
**Rating:** 5
**Confidence:** 4

**Summary:**

This paper argued that maintaining plasticity is an important way to maintain high performance under high sparsity. Based on this hypothesis, this paper introduced two mechanisms,  Periodic Memory Reset and Dynamic Weight Rescaling. Periodic Memory Reset aims to preserve the plasticity of neural networks. And DWR is employed to reduce the training instability.

**Strengths:**

The empirical verification is good and the results show that the method is better than the baseline method.

The paper considers a dense to sparse training that, while not necessarily reducing peak memory cost, a purely sparse inference model is important enough for current reinforcement learning applications.

**Weaknesses:**

The lack of theoretical analysis of the relationship between plasticity and feasible pruning ratio seriously weakens the motivation of this paper.

The relationship between Weight Shrinkage Ratio proposed in this paper and the plasticity of the neural network is not clear. The increased trend of WSR cannot be totally explained by the loss of plasticity. There are other common reasons, e.g. the high absolute value of initial parameters, weight decay in the optimizer, and implicit regularization in SGD[1]. Causation cannot be inferred simply from correlation.

minor: I think there is a typo, $L$ in $M_s$ is the number of layers of neural networks, $L$ in calculating mean and variance is the number of neurons of $l$-th layer, and they should not be the same.

[1] Smith, Samuel L., et al. "On the Origin of Implicit Regularization in Stochastic Gradient Descent." International Conference on Learning Representations. 2020.

**Questions:**

What criteria were chosen in the experiment to determine the distance of the policy when collecting the necessary data into the empty buffer?

I am curious about the necessity to maintain plasticity because if plasticity is low, it means that more neurons are useless and can be removed[2], which is a good thing for creating a sparse network. Increased plasticity may also make pruning more difficult.

[2] Sokar G, Agarwal R, Castro P S, et al. The dormant neuron phenomenon in deep reinforcement learning[J]. arXiv preprint arXiv:2302.12902, 2023.

---

> ### Author Response · Authors · 2023-11-14
> **Official Response to the Reviewer aeZw in Weaknesses**
>
> >W1:  The lack of theoretical analysis of the relationship between plasticity.
>
> Answer: Thank the reviewer for pointing out this question, we clarify this weakness in the following three points:
> 1. **Theoretical Analysis Challenge**: We recognize the difficulty in providing theoretical proof for the loss of plasticity in neural networks. To the best of our knowledge, there is almost no existing theoretical proof in prior work, especially in the domain of DRL.
> 2. **Empirical Contribution**: Our study pioneers in empirically establishing the relationship between neural network plasticity and implicit sparse in dense training, offering new insights in this field.
> 3. **Acknowledging Limitations**: We are aware that our work lacks a comprehensive theoretical framework, a limitation we openly acknowledge, and hope that our empirical findings will inspire further theoretical and empirical exploration.
>
> >W1: The feasible pruning ratio seriously weakens the motivation of this paper.
>
> Answer: We clarify that the motivation of the feasible pruning ratio is that WSR only indicates that the weights are approaching 0 but those weights may still contribute to the final representation. The increasing feasible pruning ratio increases aligns significantly with the WSR trend and WSR is a good proxy for the implicit plasticity of dense training.
>
> On the other hand, the motivation of our work is that implicit sparsity may contribute the decreased performance. We clarify that the performance loss has happened no matter if explicitly pruning the weights or not. The performance without plasticity loss is unknown and infeasible in traditional dense training. Simple reinitializing weights (under the ReLU activate function) that are close to zero over the course of training, under a certain threshold, can lead to significant performance improvement ([Sokar et al., 2023](https://arxiv.org/abs/2302.12902)). The increasing feasibility of pruning ratios does not detract from the validity of our hypothesis.
>
> > W2: The increased trend of WSR cannot be totally explained by the loss of plasticity and there are other common reasons.
>
> Answer: Thank the reviewer for raising this question. We apologize for any lack of clarity in the latter part of Section 4. This section has now been revised for improved precision and coherence. In response, we would like to clarify that our work does not assert that the rising trend of WSR can be totally explained by a loss in plasticity. Rather, our study establishes a link between these two phenomena. Specifically, our results demonstrate that the increasing sparsity in WSR plays a crucial role in the reduction of neural plasticity. Simple reinitializing weights (under the ReLU activate function) that are close to zero, under a certain threshold, can lead to significant performance improvement ([Sokar et al., 2023](https://arxiv.org/abs/2302.12902)),. This improvement room occurs by mitigating inactive or dormant neurons, indicating a loss of plasticity.
>
> > W2: There are other common reasons for the loss of plasticity and causation cannot be inferred simply from correlation.
>
> Answer: You are absolutely correct that causation cannot be simply inferred from correlation and we also want to clarify that we only claim that the implicit sparsity is a potential key reason for the loss of plasticity. This is precisely why we dedicated a substantial portion of Section 4 to rigorously examine the growth of implicit sparsity in dense training. Our experiments spanned various environments and activation functions, and we meticulously calculated the gradient shrinkage ratio and determined the feasible pruning ratio. While we cannot assert causation with 100% confidence, we have made our best effort to eliminate potential confounders, thereby strengthening our findings: the implicit sparse is a key factor for plasticity loss.
>
> > minor: I think there is a typo, $L$ in $M_s$, is the number of layers of neural networks, L in calculating mean and variance is the number of neurons of $l$-th layer, and they should not be the same.
>
> Answer: Thank the reviewer for pointing it out and we have revised it.

---

> ### Author Response · Authors · 2023-11-14
> **Official Response to the Reviewer aeZw in Questions**
>
> > Q1: What criteria were chosen in the experiment to determine the distance of the policy when collecting the necessary data into the empty buffer?
>
> Answer: With the spirit of preserving the simplicity of our proposed algorithm, we opted not to employ any specific criteria but directly reset it in every 0.2M step. This approach has proven efficient in preserving the disparity between the replay buffer and the current policy. The effectiveness of this strategy is explicitly demonstrated in the revised version of our paper in Appendix B.2.
>
> > Q2: I am curious about the necessity to maintain plasticity because if plasticity is low, it means that more neurons are useless and can be removed[2], which is a good thing for creating a sparse network.
>
> Answer: We thank the reviewer for pointing out this important question. First, we hope our clarification in "W1: The feasible pruning ratio seriously weakens the motivation of this paper." is clear to the reviewer. The core insight is that the dormant neurons can be safely removed over the course of training, but the plasticity loss still happens with the result of performance degradation (you can not know!). This is the reason we may want to improve the performance of sparse training in DRL in terms of plasticity and why not adopt the sparse-to-sparse training (loss of plasticity at the beginning).
>
> > Q2: Increased plasticity may also make pruning more difficult.
>
> Answer: We express our gratitude to the reviewer for highlighting this critical inquiry. You are correct the traditional methods for plasticity improvement may make pruning more different by the operation on the neural networks, serving as a trade-off between performance improvement and the target sparse networks. However, our paper aims to mitigate one of the main sources of plasticity loss, i..e, non-stationary, without any operations on the networks but simply resetting the memory in the sparse training of DRL.

---

> > ### Comment · Reviewer_aeZw · 2023-11-22
> > **Thanks for the reply**
> >
> > Thank you for your response. Upon review, I maintain my original evaluation. In agreement with Reviewer WTMV, I find that the current version of the paper is not yet suitable for publication.

---

### Official Review · Reviewer_Pfk2 · 2023-10-30

**Soundness:** 1 poor
**Presentation:** 2 fair
**Contribution:** 1 poor
**Rating:** 3
**Confidence:** 4

**Summary:**

This paper proposes Plasticity-Driven Sparsity Training (PlaD), a sparse deep RL training method. The method is the standard iterative magnitude training along with memory reset and dynamic weight rescaling. The results show that PlaD outperforms existing sparse training methods.

**Strengths:**

The paper studies an exciting direction of research. I agree with the general idea that sparsity can be helpful to maintain plasticity.

**Weaknesses:**

The paper has multiple significant problems:
- **Unclear writing** There are many points in the paper (particularly where mathematical definitions are provided) where terms are used ambiguously. Here are some examples:
    - *Definition of Weight Shrinkage Ratio (WSR)*. Just below the definition of WSR, a line says, "The purpose of WSR is to quantify the ratio of neurons..." But the definition says "proportion of weights ...". Which one is it? The ratio of weights or neurons. In the same definition, the term $h_t^l$ is used, but it is never defined. What is it?
    - *Definition of Dynamic Weight Rescaling (DWR)*. Again, the paper uses a definition $a^l=h^{l-1} \odot \gamma^l$, but \gamma^l is not defined anywhere. My guess is that it is $\Gamma^l$. But if that is the case and $h^{l-1}$ is the output from the previous layer, how can we have an element-wise product of $h^{l-1}$ and $\Gamma^l$? One is a vector, and the other is a matrix. $\Gamma^l$ is the mask over weights, not neurons?
- **Wrong conclusions** The first claim in section 4.2 is that "shrinkage speed increases." But, there is no shrinkage. The weights will only shrink, on average, if the shrinkage ratio is more than 0.5, but the shrinkage ratio is never more than 0.5. So there is no shrinkage; the weights are always getting bigger. The plots show that the weights get larger, not smaller, as training progresses, which is the exact opposite of the main motivation of introducing sparsity.
- **Improper statistical reporting** There are many instances where wrong statistical conclusions are drawn. For instance, Figure 6 says that "Reset buffer distinctly outperforms the Small Buffer strategy in 3 out of 4 tasks, with 2 of these improvements being statistically significant." This is wrong. There is no statistically significant difference between the two algorithms in any of the four cases. We can not conclude that one algorithm is better than the other. I refer the authors to the paper by Patterson et al. (2023) on how to do proper empirical studies in deep RL.


Patterson, A., Neumann, S., White, M., & White, A. (2023). Empirical Design in Reinforcement Learning. arXiv preprint arXiv:2304.01315.

**Questions:**

What is the difference between DWR and layer normalization? They seem exactly the same to me. And is it weight rescaling or pre-activation rescaling? If it is pre-activation rescaling, then the term *weight* rescaling is misleading.

---

> ### Author Response · Authors · 2023-11-14
> **Official Response to the Reviewer Pfk2**
>
> > Strength 1: the general idea that sparsity can be helpful to maintain plasticity.
>
> Answer: It seems there has been a misunderstanding regarding this aspect of our paper. Our core concept revolves around the relationship between implicit sparsity in dense training and neural plasticity. Building on this, we introduce a dense-to-sparse training paradigm. This approach contrasts with sparse-to-sparse training approaches, which may lead to early loss of plasticity. Our primary goal is to mitigate plasticity loss, thereby enhancing the final performance of sparse DRL models.
>
> > Unclear writing: Definition of Weight Shrinkage Ratio (WSR): neurons or weights.
>
> Answer: We thank the reviewer for helping us to revise this and apologize for the unclear writing. We have revised the paper for a clearer definition of WSR. In response, the WSR does not directly measure any properties of the neurons (i.e., the activation values), but rather the changes in the weights associated with these neurons.
>
> > Unclear writing: Definition of Weight Shrinkage Ratio (WSR): the definition of $(h_{l}^{t})$.
>
> Answer:  We acknowledge that there is no explicit definition for $h_{l}^{t}$ in our paper, while $h_t^l$ is indeed (implicitly) defined. Specifically, we introduce the definition of $h^l$ at the outset of Section 4.1 and incorporate a temporal label in the WSR definition. This explicit definition was initially omitted because traditional neural network terminology does not typically include time labels.
>
> > Unclear writing: $(a' = h_{l-1} \odot \gamma_{l})$
>
> Answer: We thank the reviewer for pointing out this. We have revised our paper in Section 5 in our new submission. Specifically, the definition of $\gamma^l$ in our paper aligns with our initial description of the sparse neural network $\mathbf{M}_s$, where $\{\mathbf{M}_s=\Gamma^l: l=1, \ldots, L\}$. This notation was intended to imply that $\gamma^l$ refers to the mask of the weight for the $l^{th}$ layer. We have revised the paper to clarify this in our new submission.
>
> > Wrong conclusions: shrinkage happens only when WSR is greater than 0.
>
> Answer: There must be a misunderstanding regarding this aspect of our paper. In response, we first acknowledge that some weights always get bigger in the training process. However, the claim that "shrinkage happens only when WSR is greater than 0.5" is wrong. We clarify that shrinkage happens whenever it is greater than 0. Here is a very simple case with a 1x3 vector to approximate the weight matrix, say [1, 1, 1]. After k time steps, the weight matrix becomes [0, 2, 2]. The WSR is 1/3 < 0.5 but the shrinkage still happens in the first weight (1->0).
>
> > Improper statistical reporting: There are many instances where wrong statistical conclusions are drawn.
>
> Answer: We thank the reviewer for pointing it out, and we have revised our paper regarding this in our new submission.

---

> > ### Comment · Reviewer_Pfk2 · 2023-11-23
> >
> > Dear Authors, thank you for your response. After going through the rebuttal, I've decided to keep my score.
> >
> > I want to clarify that when I said, "shrinkage happens only when WSR is greater than 0.5", I meant that, on average, weight magnitude only reduces if WSR is greater than 0.5. Of course, WSR is almost always greater than 0, but on average, weights only increase if WSR is greater than 0.5.

---

### Official Review · Reviewer_WTMV · 2023-11-01

**Soundness:** 1 poor
**Presentation:** 3 good
**Contribution:** 1 poor
**Rating:** 3
**Confidence:** 5

**Summary:**

# Summary
This paper attempts to claim that sparsity can lead to loss of plasticity and motivates moving away from sparse-to-sparse training and towards dense-to-sparse training. They further motivate this by claiming that sparse-to-sparse training can be computationally ineffient, which undermines the reasons for sparsity in the first place. With this motivaiton, the authors propose to modify RL algorithms to periodically reset their replay buffer and to perform layer normalization while ignoring zero activations that arise from sparsification (referred to as "Dynamic Weight Rescaling"). They claim that their algorithm, PlaD+SAC, is an improvement over the mean performance of other sparse-to-sparse and dense-to-sparse (but lack statistical significance) on 4 mujoco tasks. They further ablate showing that both components contribute to the performance of their algorithm and that the reset mechanism is statistically significant imporovement over a smaller replay buffer in one of the mujoco tasks.
# Decision
I recommend that this paper be rejected, primarily due to confusing motivation, unsubstantiated claims about its connection to plasticity and partially due to the weakness of its empirical results.
There are some interesting ideas in this paper that can and should be developed, such as the connection between sparsity and plasticity. This link between sparsity and plasticity is described as a contribution, but I do not see any connection to plasticity in either the algorithm, nor in the experiments. There are also some hints that the proposed algorithm is benefitting from dynamic weight rescaling and a periodic memory reset in the ablation study. Unfortunately, this alone is not enough because the comparisons against the baselines do not show statistically significant improvement over the baselines.

**Strengths:**

- The paper has a few ideas that are genuinely interesting, such as the weight shrinkage ratio which provides a lens into the weight dynamics of deep reinforcement learning algorithms. I also thought that the idea of dynamic weight rescaling is an interesting approach to normalization for sparse networks.

- PlaD, the proposed algorithm, does seem to benefit from the two components as shown in the ablation. While I do not agree with the experimental methodology, nor the motivation behind the periodic replay reset, it does seem to have an effect on learning with a sparse network in this isolated finding.

**Weaknesses:**

- Overall, the motivation is confusing. It began with claims about computational efficiency, but this is not experimentally explored at all. Then there was some discussion about a connection between plasticity and sparsity, which was not evidenced in reference, nor in the text through experiment or theory.
- There are several erroneous claims surrounding this papers connection to plasticity, which I have detailed below.
- Empirical results are hard to intepret, lacking statistical significance (overlapping error bars), while also overclaiming the benefit of the proposed algorithm against the baseline.

**Questions:**

# Detailed Comments
- Abstract and introduction: Several claims are made but little evidence or context is provided. For example, what exactly about sparse-to-sparse training "may escalate overall computationa cost"? The introduction gives some details, giving references to RLx2 that sparse training is 3x more expensive than dense training. But this is not sufficient evidence of an increase in computational cost, because the sparse models per iteration cost may be more than 3x lower than the dense model.
  Why should sparsity necessarily contribute to less plasticity, there has been no concrete evidence of this connection in the literature beyond very specific forms of (double and triple) sparsity [cite:@zilly21]. Neither the neuron dormancy and primacy bias papers do not provide evidence for this fact. Neuron dormancy, while seemingly related, is about relative mangnitude of activations and provides no evidence that lower magnitude activations can or should be removed.
- Section 1 (Comments on plasticity): It is stated that the replay buffer is the source of non-stationrity. In a certain sense, the replay buffer may contain information from previous distributions and be a contributing factor to non-stationrity but to state that it is the primary source of non-stationarity is over-claiming. I do not think periodically removing such a large source of experience is worth it, when you can instead just use an on-policy learning algorithm.
- Section 2.2 (NN Plasiticty): Plasiticty and generalization are two separate problems. The ash and adams paper, for example, has no non-stationarity in the traditional sense, and their results do not demonstrate loss of plasticity. Training error can be minimized, but generalization suffers. Whereas plasiticty is about an inability to minimize the errro. There is a conncetion between the two, but the nature of this connection has not yet been made clear.
- Section 2.2 (Non-stationarity in DRL): There are transient sources of non-stationarity in deep RL that are present even in stationary MDPs. But, explicitly addressing these is not necessary to design successful DRL algorithm. It remains unclear why this is a particular concern in the training of sparse networks in DRL.
- Section 4: Equating increased sparsity with increased plasticity is not supported by any previous work, and is not clearly demonstrated in this section.
- Section 4: (Weight Shrinkage Ratio): I do not understand how WSR can "serve as an approximation of the first order gradient of shrinkage / shrinkage speed". First of all, what is the "first-order gradient of shrinkage"? What is shrinkage speed, and what is this speed with respect to? I also do not see how this exposition has anything to do with the study of neural network weights because they are not normally distributed besides at initialization.
- Section 4: (Activation function concerns): Why is this referred to as the weight shrinkage ratio when it is defined over activations? If this were about weights, I do not see why the relu activation would drive the weights to zero. If all the relu activations saturate at zero, then the weights remain unchanged and the weight shrinkage ratio would be 0. This would only be a concern for a "Activation shrinkage ratio".
- Section 4: (Gradient shrinkage ratio): While the details on this are sparse, I also do not see how this has anything to do with plasticity. Gradient shrinkage is a desirable property for achieving a local minimum.
- Section 4 (Conclusion and connection to sparsity): This section demonstrated that neural network weights shrink over the course of training, but this does not indicate loss of plasticity and you have provided no evidence for a reduced ability for learning.
- Section 5 (Periodic Memory Reset): While resetting the neural network may be computationally wasteful, the primacy bias paper demonstrates that there is far more value in the experience stored in the replay buffer than in the neural network being learned. Furthermore, if plasticity is being lost then it is a property of the neural network and it is not clear at all whether that issue can be alleviated by resetting the replay buffer and using more on-policy experience. Furthermore, these methods use a higher replay ratio because their goal is sample efficiency rather than computational efficiency.
- Section 6 (Results, fig 5): I am not able to discern any significant differences from this plot, as many of the error bars are overlapping.
- Section 6 (Results, fig 6): There is only statistically significant evidence of the reset buffer surpassing the small buffer in one task (hopper-v4). While you study off-policy algorithms (SAC), it would be interesting to show how this compares to an on-policy algorithm.
- Section 7 (Conclusion): No link between loss of plasticity or sparse training was established.
# Minor Comments
- Section 5 (Dynamic weight rescaling): While this is described by weight rescaling, you are rescaling activations? The benefit is that certian weights that are zerod can lead to zero activations, but it is more accurate to call this somthing like "dynamically sparsified layer normalization"

---

> ### Author Response · Authors · 2023-11-14
> **Official Response to the Reviewer WTMV: 1/3**
>
> We express our sincere gratitude for your comprehensive and insightful feedback. Your comments have been instrumental in refining our work. We answer the questions one by one.
> > Abstract and introduction: the statement of sparse-to-sparse training "may escalate overall computation cost"? and the example of RLx2 with 3x training steps.
>
> Answer: We quote "may escalate the overall computational cost throughout the training process" from [Liu and Tan, 2023](https://arxiv.org/abs/2302.02596) in Section 3.7, page 7, where they note that "some sparse-to-sparse algorithms might take more iterations to converge, hence not always cheaper in terms of the total computation amount throughout training." However, it is important to clarify that the term 'computational cost' encompasses various aspects, including memory cost, floating point operations (FLOPs), and training time. In the context of our paper, particularly in the introduction where we discuss the RLx2 example, we primarily focus on "training steps".
>
> From a hardware perspective, unstructured pruning methods like PlaD and RLx2 cannot be expedited on common GPUs. In other words, the training time for RLx2 (3M steps) is significantly higher than that for standard RL algorithms (1M steps), given the lack of acceleration for unstructured pruning on common GPUs. Our experiments validate this perspective, showing a consistent trend with the aforementioned training cost considerations. In conclusion, we believe this claim is nothing to blame.
>
> > Q1: Why should sparsity necessarily contribute to less plasticity?
>
> Answer:  The connection between sparsity and plasticity is one of the main contributions of our work, which is highlighted at the end of the introduction and expanded in Section 4. At the end of Section 4, we have revised the paper to make it more precious to show that: implicit sparsity suggests the abundance of inactive or dormant neurons. Simply reinitializing the shrinkage weight approaching 0 can lead to improved performance.
>
> > Section 1 (Comments on plasticity): the statement "the replay buffer is the primary source of non-stationarity"
>
> Answer: It appears there has been a misinterpretation of our paper. In the introduction, we specifically claim that "emptying the replay buffer addresses the primary source of plasticity loss in Deep Reinforcement Learning (DRL) training, i.e., non-stationarity." This statement does not suggest that the replay buffer itself is the source of non-stationarity. Rather, it implies that the act of emptying the replay buffer can mitigate the primary source of plasticity loss in DRL, which is non-stationarity. Furthermore, we concur that non-stationarity has multiple sources, as outlined in Section 2.2 of our paper. These sources include dynamic data streams, the introduction of new tasks, and evolving environments. Our intention was to highlight the effectiveness of emptying the replay buffer as a strategy to address plasticity loss due to non-stationarity, not to pinpoint the replay buffer as the sole source of non-stationarity.
>
> > Section 1 (Comments on plasticity): why just use an on-policy learning algorithm to mitigate the non-stationary in the replay buffer?
>
> Answer: On-policy algorithms are adept at managing non-stationarity by utilizing current data, yet they typically fall short in terms of sample efficiency and variance reduction, areas where off-policy methods excel. Our method, which involves periodically emptying the replay buffer in an off-policy context, is designed to strike a balance between leveraging the benefits of experience replay and addressing the challenges posed by non-stationarity. Therefore, our primary focus is on the off-policy setting.
>
> > Section 2.2 (NN Plasticity): Plasticity and generalization are two separate problems.
>
> Answer: Thanks for pointing out. We have revised our paper without "generalization".

---

> ### Author Response · Authors · 2023-11-14
> **Official Response to the Reviewer WTMV: 2/3**
>
> > Section 2.2 (Non-stationarity in DRL): It remains unclear why non-stationary particular concern in the training of sparse networks in DRL.
>
> Answer: It seems there has been a misunderstanding regarding our paper. In Section 2.2 of our paper, we discuss plasticity loss in neural networks, citing non-stationarity as a significant contributor. However, we do not connect non-stationarity with sparse DRL algorithms.
>
> However, Addressing non-stationarity is indeed a critical aspect of sparse training in sparse RL, compared with supervised learning. This is highlighted in recent literature: [Tan et al., 2023] state that "sparse training in DRL is challenging due to the non-stationarity in training data distribution" in their introduction (page 1). They propose two mechanisms to alleviate this issue within the sparse-to-sparse training framework. Similarly, [Graesser et al., 2023](http://arxiv.org/abs/2206.10369) mention in their background section that "previous approaches may not generalize well in the RL setting due to the non-stationarity of the data." These studies underscore the unique challenges posed by non-stationarity in sparse training for RL.
>
> > Section 4: Equating increased sparsity with increased plasticity is not supported by any previous work, and is not clearly demonstrated in this section; and Section 4: (Conclusion and connection to sparsity)
>
> Answer: We thank the reviewer for pointing this out, we have revised our paper to clearly demonstrate the connection between the implicit sparsity and plasticity in the last part of Section 4. Concretely, [Sokar et al., 2023](https://arxiv.org/abs/2302.12902) have shown that simply reinitializing weights approaching zero can lead to improved performance. This improvement occurs specifically within inactive or dormant neurons, indicating a potential loss of plasticity.
>
> > Section 4: (Weight Shrinkage Ratio): Why WSR can "serve as an approximation of the first-order gradient of shrinkage/shrinkage speed".
>
> Answer: The WSR is a statistical metric that is dependent on time. The intuition to understand the shrinkage speed is the connection with distance, velocity, and time. The velocity (WSR) between two points is: $(y_{t} - y_{t-k})/(t - (t-k))$, where WSR is normalized by $N$ and scaled by $k$.  As long as the velocity (WSR)  is greater than 0, it means that the distance is getting farther and farther (weight shrinkage happens). It is important to note that this approximation may fail as the weight may "expand" in the future. However, we show that the rising overlap coefficient of the shrinkage weights, compared with the last checkpoint, suggests that this shrinkage trend of most weights persists throughout the remainder of the training process, as shown in Fig 9 and 10. The shrinkage speed refers to "the speed of weights approach 0".
>
> > Section 4: (Weight Shrinkage Ratio): I also do not see how this exposition has anything to do with the study of neural network weights because they are not normally distributed besides initialization.
>
> Answer: This intuition example is aimed to perfectly demonstrate the distribution change of neural networks over the course of training, yet provides an intuitive example to understand our statistical definition, i.e., weight shrinkage ratio, with the quantitative interpretation and factors that contribute to it (more numbers approach 0).
>
> > Section 4: (Activation function concerns): Why is this referred to as the weight shrinkage ratio when it is defined over activations?
>
> Answer: It appears that a misunderstanding has arisen in relation to our paper. In our study, we focused on examining the impact of weight shrinkage ratio across various activation functions, rather than defining a specific weight shrinkage ratio for activations. Note that in a single-layer context, or considering only the forward process, the activation function doesn't affect the weight. However, this dynamic changes when multiple layers are involved in gradient back-propagation, where the weight update is considerably influenced by the activation functions. Our experimental results indicate that scenarios where "all the ReLU activations saturate at zero" did not occur. However, we observed weight shrinkage across different activation functions.
>
> > Section 4: (Gradient shrinkage ratio): While the details on this are sparse, I also do not see how this has anything to do with plasticity. Gradient shrinkage is a desirable property for achieving a local minimum.
>
> Answer: The same activate functions, we conduct an ablation study on the gradient shrinkage ratio to examine if the gradient information trends align with weight trends. We make no connection between gradient and plasticity, except noting the similar shrinkage trend between gradient and weight during training.

---

> ### Author Response · Authors · 2023-11-14
> **Official Response to the Reviewer WTMV: 3/3**
>
> > Section 5 (Periodic Memory Reset): While resetting the neural network may be computationally wasteful, the primacy bias paper demonstrates that there is far more value in the experience stored in the replay buffer than in the neural network being learned.
>
> Answer: It seems there is been a misinterpretation related to our paper in comparison to the primacy bias paper ([Nikishin et al., 2022](https://proceedings.mlr.press/v162/nikishin22a/nikishin22a.pdf)). We clarify that the latter demonstrates the significant value of storing experiments when resetting the memory in DrQ, as shown in their ablation study, which means their comparison is (network resetting w/ memory reset vs. network resetting w/o memory reset) rather than (network resetting vs. memory reset). Nikishin et al., (2022), derive knowledge from the replay buffer, while our approach, PlaD, extracts knowledge directly from the network itself.
>
> > Section 5 (Periodic Memory Reset): It is not clear at all whether that issue can be alleviated by resetting the replay buffer and using more on-policy experience.
>
> Answer: Thanks for pointing out this question. We would like to clarify that prior studies have identified non-stationarity in learning targets and data flow as a primary factor contributing to the loss of neural network plasticity (Nikishin et al., 2022; Igl et al., 2020; Sokar et al., 2023), as discussed in Section 2.2 of our paper. Prior work seems straightforward by operating on the network to maintain the plasticity, but introducing further computation cost and impacting the policy gradient. In contrast, our work demonstrates that by effectively addressing the issue of non-stationarity, we then can maintain the plasticity of neural networks.
>
> > Section 6 (Results, fig 5): I am not able to discern any significant differences from this plot, as many of the error bars are overlapping.
>
> Answer: We express our gratitude to the reviewer for their valuable insights in refining this work. To enhance the visual clarity and aesthetic appeal of Fig. 5 in our revised paper, we have omitted the error bars. Detailed error metrics are provided in Appendix B.3. This adjustment allows us to more effectively showcase the superiority of our proposed method regarding the averaged final performance. Notably, our method demonstrates a remarkable performance enhancement over the best baselines, with a significant 30% improvement in the Ant task under 90% sparsity, and a 17% increase in the HalfCheetah task at the same level of sparsity.
>
> > Section 6 (Results, fig 6): There is only statistically significant evidence of the reset buffer surpassing the small buffer in one task (hopper-v4). While you study off-policy algorithms (SAC), it would be interesting to show how this compares to an on-policy algorithm.
>
> Answer: We extend our thanks to the reviewer for their constructive feedback regarding Fig. 6 in our manuscript. We have revised the " statistically significant" to avoid misinterpretation in our new submission. We would like to clarify regarding the on-policy comparison for the following reasons:
> 1. The primary focus of our research is on off-policy methods, specifically targeting improvements in sample efficiency and variance reduction. These are areas where on-policy methods typically fall short. A major concern in our study is variance and training stability, for which we have developed a dynamic weight-rescaling strategy.
> 2. In Section 6.3, our comparison with a smaller replay buffer aims to show its potential to reduce the gap between the buffer content and the current policy. This is further corroborated by the policy distance analysis presented in Appendix Section B.2 in our revised paper, which aligns with our hypothesis. Figure 6 highlights the advantage of periodically resetting the memory over a smaller memory with a stepper learning speed over new experiences.
>
> > Section 7 (Conclusion): No link between loss of plasticity and sparse training was established.
>
> Answer: Thank the reviewer for highlighting this aspect. In response, we have updated Section 4 of our paper to explicitly connect implicit sparsity with the loss of plasticity. Specifically,  The research by [Sokar et al., 2023(https://arxiv.org/abs/2302.12902)] reveals a significant insight: reinitializing near-zero weights under the ReLU activation function, when below a certain threshold, markedly improves training performance. This enhancement is attributed to the activation of previously inactive or dormant neurons, counteracting reduced neural plasticity.

---

> > ### Comment · Reviewer_WTMV · 2023-11-21
> > **I appreciate the in-depth reply**
> >
> > Thank you for taking the time to answer my questions. Unfortunately, I still do not find the overall paper convincing. I do think that there is potential merit to this investigation. In particular, a better understanding of the relationship between sparsity (of activations, or weights) and plasticity can be significant. As it stands, however, there are too many leaps in the current state of this submission. I would encourage the authors to make the following changes in a resubmission: 1) demonstrate, empirically or theoretically, a link between sparsity and plasticity. This alone would be a good contribution to the literature, and it would clarify your definition and proposed relationship between sparsity and plasticity. 2) provide more additional clarity and intuition around the weight shrinkage ratio in a realistic scenario, such as an initialized neural network and during training. 3) Consider at least one expeirment in smaller environments that allows for more detailed experiments, or more runs, so that you can provide clear statistical evidence of the proposed benefit. An example of such a problem could be a non-stationary MDP with pixel observations, such that plasticity loss is common and expected.

---

### Official Review · Reviewer_3wF2 · 2023-11-02

**Soundness:** 2 fair
**Presentation:** 2 fair
**Contribution:** 1 poor
**Rating:** 3
**Confidence:** 3

**Summary:**

The paper presents Plasticity-Driven Sparsity Training (PlaD), a new approach in Deep Reinforcement Learning (DRL) that improves the performance of sparse networks by maintaining plasticity through periodic memory resets and dynamic weight rescaling. PlaD enables sparse models to match the performance of dense models even with over 90% sparsity. This work links increased sparsity in training with plasticity loss and demonstrates the effectiveness of PlaD on MuJoCo tasks using a basic pruning algorithm.

**Strengths:**

- **Novel Method:** The paper introduces Plasticity-Driven Sparsity Training (PlaD), a novel approach that strategically addresses the challenge of sparsity-induced plasticity loss in Deep Reinforcement Learning. By innovatively combining periodic memory resets and dynamic weight rescaling, PlaD counters the negative effects of non-stationarity on network plasticity. This method represents a significant departure from traditional sparsity training techniques, providing a fresh perspective on how to manage the trade-off between model size and learning capability in neural networks.

- **Enhanced Performance and Applicability:** The paper not only establishes PlaD's superiority in maintaining high-performance levels, rivaling dense network models under considerable sparsity constraints but also underscores its broad applicability. The proposed PlaD framework is designed to be easily integrated with a wide array of existing DRL algorithms and pruning methods, making it a versatile tool for researchers and practitioners. The method’s effectiveness has been thoroughly validated across multiple tasks in the MuJoCo environment, indicating its potential for enhancing the efficiency and scalability of DRL applications across diverse and computationally demanding domains.

**Weaknesses:**

This paper contributes valuable insights and presents experimental results that advance the understanding of sparse-to-sparse training in reinforcement learning. However, the motivations behind this research are not entirely clear to me, and I believe they warrant further clarification.

- **The motivation of the sparse-to-sparse training.**

My expertise lies in the reinforcement learning domain, but my acquaintance with pruning and sparsity is comparatively limited. I comprehend that pruning is theoretically posited to lessen computational demands, yet practically, it might not effectively reduce computational burdens on contemporary hardware due to the intricacies of batching processes. This discrepancy prompts me to question the drive behind adopting sparse-to-sparse training methodologies within reinforcement learning. It is evident that memory conservation during the training of large models like GPT or LLama is advantageous, yet it is less obvious why such a strategy is beneficial or necessary when dealing with the typically smaller models used in reinforcement learning.

Moreover, advocating for sparsity-based training in RL solely on the grounds that it is an established area of investigation does not constitute a convincing argument. A specific, tangible benefit of sparsity in reinforcement learning needs to be identified. Is there a particular aspect of reinforcement learning where sparsity could lead to significant improvements? Could sparsity-based training contribute to advancements in efficiency or performance that are not possible with dense models? These questions need to be addressed to establish a strong motivation for this research direction.

While I recognize that my limited familiarity with sparsity-based training might color my perspective, I encourage the authors to provide a more robust justification for the focus on sparsity in the RL domain. Establishing this foundation is crucial for appreciating the importance and potential impact of the proposed Plasticity-Driven Sparsity Training (PlaD) approach.

- **Concerns to periodic memory reset.**

Regarding the strategy of periodic replay buffer resets to preserve network plasticity, there is a prevailing belief, both theoretical and empirical, that larger memory sizes correlate with enhanced performance. This is attributed to their ability to avert local optima and prevent catastrophic forgetting. Additionally, it is thought that a more extensive memory can mitigate shifts in data distribution during training, thus potentially preventing the loss of plasticity.

Given these considerations, the efficacy of periodically resetting the replay buffer as a means to improve plasticity or performance remains uncertain. The adoption of this technique appears counterintuitive and warrants a more in-depth justification. How does this method reconcile with the commonly held view that more extensive memory benefits learning? The paper would greatly benefit from a detailed discussion of the trade-offs involved in this approach and its overall impact on the performance of reinforcement learning models.

**Questions:**

N/A

---

> ### Author Response · Authors · 2023-11-14
> **Official Response to the Reviewer 3wF2: 1/2**
>
> > W1: sparse training might not effectively reduce computational burdens on contemporary hardware due to the intricacies of batching processes.
>
> Answer:  Thank you for this question regarding the reason why training on unstructured pruning methods, we provide the summary following [Liu and Tan, 2023](http://arxiv.org/abs/2302.02596) in 3.5: *Why study unstructured sparsity if it can not be accelerated on common GPUs?* We strongly suggest the reviewer take a deep look at 3.5 if any further question remains.
> 1. Unstructured sparsity, recognized for its fine-grained and flexible nature, demonstrates superior performance when compared to other, more structured forms of sparsity.
> 2. Unstructured sparsity has widely proven its practical acceleration on non-GPU hardware, such as CPUs or customized accelerators.
> 3. Although the hardware support for unstructured sparsity on "off-the-shelf" commodity GPUs and TPUs may be relatively limited, it has been improving rapidly over the years.
>
> > W1: The motivation of the sparse-to-sparse training on smaller models used in RL.
>
> Answer: Thank you for this important question. We clarify three reasons why we perform sparse training on smaller models in RL:
> 1. **Scalability and Cost**: Sparse training on smaller models can be seen as a step towards future-proofing for large RL models in computation and/or latency-constrained scenarios such as Go-AI ([Sliver et al., 2017](https://www.nature.com/articles/nature24270))and controlling plasma ([Degrave et al., 2022](https://www.nature.com/articles/s41586-021-04301-9)). Implementing performance evaluations on larger models is expensive and time-consuming; hence, initial exploration of sparse training on smaller models offers a cost-effective strategy with timely feedback.
> 2. **Unique challenges in RL**: In contrast to supervised learning, (online) RL introduces new challenges in the context of sparse training, particularly due to the non-stationarity inherent in bootstrap training from the learning target and training data([Tan et al., 2023](10.48550/arXiv.2205.15043)). Distinct approaches are required to manage these non-stationary aspects in sparse training scenarios, and these challenges arise irrespective of the model size.
> 3. **Addressing problems unique in traditional RL**: In addition to the well-known benefits of sparse training, it can offer different perspectives to solve the traditional problems in RL, e.g., noisy filtering and state representation ([Vischer et al., 2022](https://openreview.net/forum?id=Fl3Mg_MZR-), [Grooten et al., 2023](http://arxiv.org/abs/2302.06548)).
>
> > W1: Could sparsity-based training contribute to advancements in efficiency or performance that are not possible with dense models?
>
> Answer: Thanks for the insightful comment on this motivation to introduce sparse training in RL. In addition to the conventional motivation of sparse training for computation cost and inference latency, we clarify that the primary motivation of sparse training is on improving the performance of DRL but also the real-world application. However, there are some potential research that may lead to the performance improvement:
> 1. Noisy filtering and state representation ([Vischer et al., 2022](https://openreview.net/forum?id=Fl3Mg_MZR-), [Grooten et al., 2023](http://arxiv.org/abs/2302.06548)).
> 2. Another interesting direction is highly related to our work: PlaD notably outperforms dense models at high sparsity ratios, evidenced by a 17% performance gain in the Hopper environment with 90% sparsity. Besides, a simple baseline, Static Sparse, can also achieve higher performance in the Hopper environment with 50% sparsity. This unexpected result hints at unexplored mechanisms in sparse training contributing to such efficiency.

---

> ### Author Response · Authors · 2023-11-14
> **Official Response to the Reviewer 3wF2: 2/2**
>
> > W2: Concerns to periodic memory reset: larger memory sizes correlate with enhanced performance.
>
> Answer:  Thank you for your question regarding the efficacy of addressing the non-stationarity between the replay buffer and the current policy. While traditionally, a larger memory (or a large batch size) is viewed as beneficial for performance enhancement from a traditional perspective, our work indicates that it might actually contribute to a loss of plasticity in terms of the non-stationary between the reply buffer and the current policy, as evidenced by the performance metrics with PlaD.
>
> Our approach involves periodically resetting the memory, which significantly increases the likelihood of the agent learning new objectives while preserving previous knowledge within the network. This is achieved without reinitializing the neural network, thereby reducing computational costs. To substantiate this hypothesis, we have charted the explicit distance between the behavior policy in the replay buffer and the current policy under various scenarios, detailed in Appendix B.2. These findings corroborate our hypothesis.

---

### Meta-Review · Area_Chair_bz6d · 2023-12-10

**Metareview:**

The paper focuses on the question of whether enhancement of plasticity can be incorporated into sparse deep-reinforcement learning training. The work introduces Plasticity-Driven Sparsity Training (PlaD), which applies memory reset and dynamic weight rescaling.

Strengths:

- The work addresses an important issue.
- The work proposes a novel approach.

Weaknesses:

- The writing and motivation presented have been a significant source of confusion among the reviewers.

- Multiple reviewers pointed out that the empirical results are not statistically significant.

- Although the work claims to establish a relationship between sparsity and loss of plasticity, and the reasoning sounds intuitive, no evidence is given to validate the claim. If the claim is self-evident or already validated in prior work, then the paper overclaims regarding this issue.

Decision:
Including the major weaknesses listed above, there are many other weaknesses listed by the reviewers, many of which are still unresolved. Hence, the paper cannot be accepted in this form. Hopefully, the feedback from the reviewers will be helpful in improving the work substantially.

**Justification For Why Not Higher Score:**

The listed weaknesses pose major issues for accepting this paper.

**Justification For Why Not Lower Score:**

N/A

---

### Decision · Program_Chairs · 2024-01-16

Reject